# Miniaturized 3D bone marrow tissue model to assess response to Thrombopoietin-receptor agonists in patients

Christian A Di Buduo[1], Pierre-Alexandre Laurent[1†], Carlo Zaninetti[2†], Larissa Lordier[3], Paolo M Soprano[1], Aikaterini Ntai[4,5], Serena Barozzi[2], Alberto La Spada[4,5], Ida Biunno[5,6], Hana Raslova[3], James B Bussel[7], David L Kaplan[8], Carlo L Balduini[2], Alessandro Pecci[2], Alessandra Balduini[1,8]*

[1]Department of Molecular Medicine, University of Pavia, Pavia, Italy; [2]Department of Internal Medicine, I.R.C.C.S. San Matteo Foundation and the University of Pavia, Pavia, Italy; [3]UMR 1170, Institut National de la Santé et de la Recherche Médicale, Univ. Paris-Sud, Université Paris-Saclay, Gustave Roussy Cancer Campus, Equipe Labellisée Ligue Nationale Contre le Cancer, Villejuif, France; [4]Integrated Systems Engineering, Milano, Italy; [5]Isenet Biobanking, Milano, Italy; [6]Institute for Genetic and Biomedical Research-CNR, Milano, Italy; [7]Department of Pediatrics, Weill Cornell Medicine, New York, United States; [8]Department of Biomedical Engineering, Tufts University, Medford, United States

**Abstract** Thrombocytopenic disorders have been treated with the Thrombopoietin-receptor agonist Eltrombopag. Patients with the same apparent form of thrombocytopenia may respond differently to the treatment. We describe a miniaturized bone marrow tissue model that provides a screening bioreactor for personalized, pre-treatment response prediction to Eltrombopag for individual patients. Using silk fibroin, a 3D bone marrow niche was developed that reproduces platelet biogenesis. Hematopoietic progenitors were isolated from a small amount of peripheral blood of patients with mutations in *ANKRD26* and *MYH9* genes, who had previously received Eltrombopag. The ex vivo response was strongly correlated with the in vivo platelet response. Induced Pluripotent Stem Cells (iPSCs) from one patient with mutated *MYH9* differentiated into functional megakaryocytes that responded to Eltrombopag. Combining patient-derived cells and iPSCs with the 3D bone marrow model technology allows having a reproducible system for studying drug mechanisms and for individualized, pre-treatment selection of effective therapy in Inherited Thrombocytopenias.

*For correspondence:
alessandra.balduini@unipv.it

†These authors contributed equally to this work

## Introduction

Bone marrow megakaryocytes are responsible for the continuous production of platelets in the blood, driven by Thrombopoietin (TPO) through interaction with its receptor MPL (*Hitchcock and Kaushansky, 2014*; *Kaushansky, 2015*). In vivo, megakaryocytes associate with bone marrow microvasculature, where they extend proplatelets that protrude through the vascular endothelium into the lumen and release platelets into the bloodstream (*Ito et al., 2018*; *Junt et al., 2007*).

Countless human pathologies result in alterations in platelet production; yet, for many of these, pathogenesis, and thus optimal targeted therapies, remain unknown. Inherited Thrombocytopenias are a diverse group of disorders characterized by low platelet count, resulting in impaired hemostasis. While often stable, patients can experience hemorrhages and/or excessive bleeding provoked

**eLife digest** Platelets are tiny cell fragments essential for blood to clot. They are created and released into the bloodstream by megakaryocytes, giant cells that live in the bone marrow. In certain genetic diseases, such as Inherited Thrombocytopenia, the bone marrow fails to produce enough platelets: this leaves patients extremely susceptible to bruising, bleeding, and poor clotting after an injury or surgery.

Certain patients with Inherited Thrombocytopenia respond well to treatments designed to boost platelet production, but others do not. Why these differences exist could be investigated by designing new test systems that recreate the form and function of bone marrow in the laboratory. However, it is challenging to build the complex and poorly understood bone marrow environment outside of the body.

Here, Di Buduo et al. have developed an artificial three-dimensional miniature organ bioreactor system that recreates the key features of bone marrow. In this system, megakaryocytes were grown from patient blood samples, and hooked up to a tissue scaffold made of silk. The cells were able to grow as if they were in their normal environment, and they could shed platelets into an artificial bloodstream. After treating megakaryocytes with drugs to stimulate platelet production, Di Buduo et al. found that the number of platelets recovered from the bioreactor could accurately predict which patients would respond to these drugs in the clinic.

This new test system enables researchers to predict how a patient will respond to treatment, and to tailor therapy options to each individual. This technology could also be used to test new drugs for Inherited Thrombocytopenias and other blood-related diseases; if scaled-up, it could also, one day, generate large quantities of lab-grown blood cells for transfusion.

by hemostatic events such as trauma or surgery; in some cases, hemorrhages appear spontaneously (*Balduini et al., 2018*; *Balduini et al., 2017*). The treatment of Inherited Thrombocytopenias is still unsatisfactory. For patients affected with the severe forms, which are usually fatal at young ages, the treatment of choice is hematopoietic stem cell transplantation (*Balduini et al., 2013*; *Locatelli et al., 2003*; *Notarangelo et al., 2008*). However, for most patients with Inherited Thrombocytopenias, transplantation is not recommended as the risks outweigh the benefits. The standard treatment protocols for these subjects were platelet transfusions to stop or prevent bleeding following trauma or during invasive procedures, anti-fibrinolytic agents, recombinant factor VIIa (rVIIa), or local treatment. A significant advance in the treatment of thrombocytopenias is the use of drugs that stimulate platelet production by mimicking the effects of TPO. The TPO-receptor agonists Eltrombopag, Romiplostim, and very recently Avatrombopag, have been approved for the treatment of several forms of acquired thrombocytopenia (*Bussel, 2018*; *Cheng, 2011*; *Erickson-Miller et al., 2009*; *Kuter, 2013*; *Santini and Fenaux, 2015*). TPO-receptor agonists were first explored in Inherited Thrombocytopenias in 2010 in a phase 2 trial of Eltrombopag in 12 patients with Myosin Heavy Chain 9 (MYH9) mutations (*Pecci et al., 2010*). In 2015, Eltrombopag was tested in eight patients with Wiskott-Aldrich syndrome with platelet increases primarily in the X-Linked Thrombocytopenia (XLT) patients (*Gerrits et al., 2015*). More recently, a follow on phase 2 trial showed that Eltrombopag was safe and effective in increasing platelet count and reducing bleeding symptoms in patients with different forms of Inherited Thrombocytopenia, including *MYH9*-Related Diseases (*MYH9*-RD), *Ankyrin Repeat Domain 26*-Related Thrombocytopenia (*ANKRD26*-RT), XLT/Wiskott-Aldrich syndrome, monoallelic Bernard-Soulier syndrome and *Integrin beta 3* (*ITGB3*)-Related Thrombocytopenia (*Zaninetti et al., 2020*). Further, elective surgeries in *MYH9*-RD patients with severe thrombocytopenia have been performed safely after administration of Eltrombopag (*Zaninetti et al., 2019*). Overall, these studies indicated that a sizeable proportion of patients with Inherited Thrombocytopenia respond to Eltrombopag, but that the extent of platelet response is highly variable not only among different forms of Inherited Thrombocytopenia but also among different patients affected by the same disease.

Tools that recapitulate the function of specific tissues or organs are critical to test drug efficacy, reduce ineffective or suboptimal therapies, and personalize the choice of the best treatment for each specific patient as exemplified by organoids. Reproduction of the bone marrow has been very

difficult because of its incompletely understood complexity. Current research is focused on duplicating characteristic features of the physiologic bone marrow microenvironment ex vivo using relevant biomaterials and bioreactors, along with appropriate human cell sources (*Chou et al., 2020*; *Di Buduo et al., 2018*; *Di Buduo et al., 2021*). Silk is a naturally derived protein biomaterial with utility for studying platelet production since its fundamental features include non-thrombogenicity, low-immunogenicity, and non-toxicity (*Abbonante et al., 2017*; *Di Buduo et al., 2017*; *Di Buduo et al., 2015*; *Omenetto and Kaplan, 2010*). A combination of modular flow chambers and vascular silk tubes and sponges was used to record platelet generation by primary human megakaryocytes, in response to variations in surface stiffness, functionalization with extracellular matrix components, and co-culture with endothelial cells (*Di Buduo et al., 2017*; *Di Buduo et al., 2015*). These systems were able to support efficient platelet formation and, upon perfusion, recovery of functional platelets, as assessed through multiple activation tests, including participation in clot formation and thrombus formation under flow conditions (*Di Buduo et al., 2017*; *Di Buduo et al., 2015*).

We developed an ex vivo miniaturized 3D bone marrow tissue model that recapitulates ex vivo platelet biogenesis of patients with different forms of Inherited Thrombocytopenias. This device is a radical improvement of the previous model because it minimizes the number of cultured cells required in an unlimited number of simultaneous culture chambers. The results, starting from only 15 mL of peripheral blood, showed that the ex vivo tissue model could predict the in vivo clinical platelet response to Eltrombopag in individual patients. The number of platelets recovered in the ex vivo model under standardized conditions, including exposure to Eltrombopag, was significantly correlated with the increase in platelet count observed in vivo after Eltrombopag treatment in the same patients. Overall, our data suggest this tissue model will have substantial applicability for the evaluation of the effects of compounds to determine their impact on platelet production.

## Results

### Device design and prototyping

In adults, hematopoietic bone marrow is located in the medullary cavity of flat and long bones (*Travlos, 2006*), served by blood vessels that branch out into millions of small thin-walled arterioles and sinusoids allowing mature blood cells to enter the bloodstream (*Figure 1A*). To mimic such a structure, a device prototype of rectangular shape with 30 × 30 × 14 mm size and hollow cavities of 2 × 15 × 3.5 mm was developed. The device was connected to an outside peristaltic electronic pump (*Figure 1A*) through 0.9 mm diameter channels equipped with luer lock adaptors. We used devices with up to two reservoirs; however, the chamber can be designed to provide as many channels as required by the experimental conditions (*Figure 1—figure supplement 1*). Crosstalk between channels inside the device was eliminated by appropriate spatial separation and independent perfusion to allow assessment of patient-specific responses, following simultaneous exposure to TPO alone and TPO in combination with the tested drug.

3D printing technology is one emerging option for producing new devices in a customized, fast, and cost-effective manner. The printing process for the negative mold of our device is easily scalable. It can be created in less than 1 hr using a polylactic acid (PLA High temperature, FormFutura Volcano, *Figure 1—figure supplement 2*), which allows casting and curing of polydimethylsiloxane (PDMS), a non-toxic polymeric organosilicon. The final shape of the system is optically clear (*Figure 1B–E*). Importantly, the device is reusable and autoclavable to ensure overall sterility to the system.

### Silk biomaterials for bone marrow system assembly and characterization

A silk fibroin structure functionalized with fibronectin was prepared with salt leaching method and inserted into the device to model a spongy scaffold that reproduces bone marrow architecture, composition, and microcirculation (*Figure 2A–C*). A 2 days production process allowed us to obtain a sterile 3D silk-fibronectin scaffold that could be stored in water, at 4°C, up to 1 month after preparation and used upon experimental needs. The silk scaffold was connected to gas-permeable tubing allowing perfusion of the media with a peristaltic pump connected to inlet and outlet ports (*Figure 2A*). A cover cap closes the system before starting perfusion. The 3D reconstruction of the

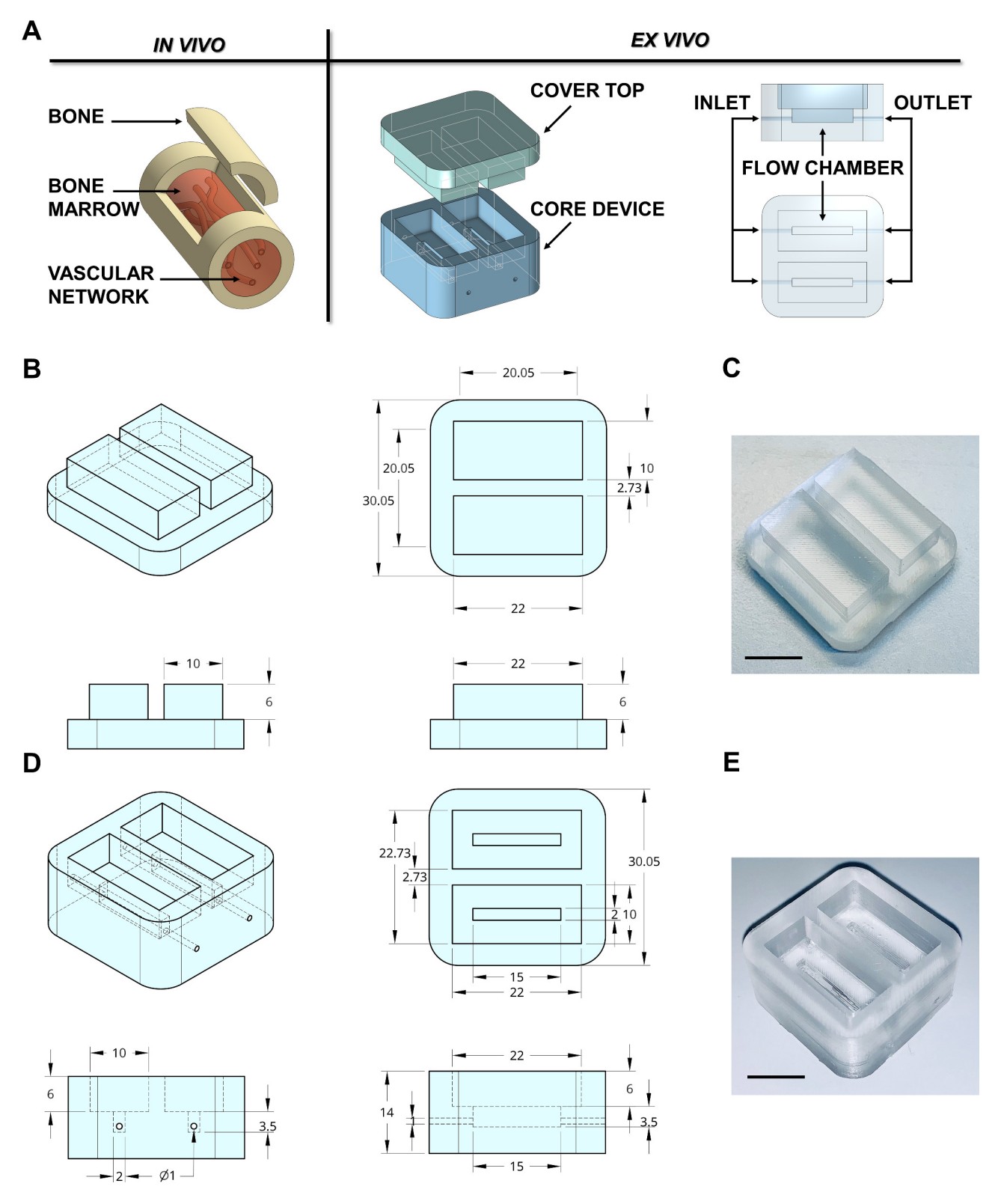

**Figure 1.** Design of the bone marrow mimicking device. (A) To mimic the vascularized bone marrow tissue structure ex vivo a double-flow chamber device was designed in two parts. The core contains two separates flow channels dedicated to the perfusion having inlet and outlet ports for connection to a perfusion system. (B,C) The dimension of the polydimethylsiloxane (PDMS) mold cover top and (D,E) the core device is expressed in

*Figure 1 continued on next page*

*Figure 1 continued*

millimeters. Alternative models of the device are shown in *Figure 1—figure supplement 1*. The 3D-printed negative mold of the chamber is shown in *Figure 1—figure supplement 2*.

The online version of this article includes the following figure supplement(s) for figure 1:

**Figure supplement 1.** Various models of the silk bone marrow device.
**Figure supplement 2.** 3D printing of the negative mold.

silk scaffold revealed the presence of multiple spatially distinct niches (*Figure 2D and E*) and also demonstrated the homogeneous distribution of pores from top to bottom of the scaffold (*Figure 2F*). This arrangement efficiently supported the diffusion of cells (*Figure 2G*) and media out-flow without altering the shape and integrity of the silk. Importantly, the total volume collected after perfusion corresponded to that injected in the system by the pump.

## Tuning of the silk bone marrow device for testing hematopoietic progenitor response to drugs

To ascertain the ability of the device to model physiological and pathological bone marrow, we took advantage of our expertise in culturing human hematopoietic stem and progenitor cells from periph-eral blood of healthy controls and patients affected by two forms of Inherited Thrombocytopenia: *ANKRD26*-RT and *MYH9*-RD (*Bluteau et al., 2014*; *Pecci et al., 2009*). The bone marrow device was able to support efficient differentiation of mature megakaryocytes from both healthy controls and patients (*Figure 3A and B*). However patient-derived megakaryocytes displayed a decreased per-centage of proplatelet formation by about 80%, accompanied by less branching of proplatelet shafts due to a significantly lower number of bifurcations (Healthy Control: $9 \pm 2$; *ANKRD26*-RT: $1.9 \pm 0.7$; *MYH9*-RD $1.8 \pm 0.9$) as compared to healthy controls (*Figure 3C–E*).

To validate the predictive value of the miniaturized bone marrow response to drugs specifically targeting hematopoiesis, we chose Eltrombopag as te model compound since Eltrombopag repre-sents to date the only tested drug shown to increase platelet count of patients with Inherited Throm-bocytopenias. First, we verified the ex vivo efficacy of Eltrombopag on human adult megakaryocytic progenitors from healthy controls and demonstrated the ability of the drug to increase megakaryo-cyte output and proplatelet formation with respect to the untreated control (*Figure 3—figure sup-plement 1*). Then, we tested the sensitivity of 24 pathological samples from *ANKRD26*-RT and *MYH9*-RD patients (*Table 1*). This cohort included 11 patients previously treated with Eltrombopag in a recent phase 2 clinical trial (*Zaninetti et al., 2020*) and two patients previously treated in prepa-ration for elective surgery (*Zaninetti et al., 2019*). Blood samples for this study were collected when patients were out of Eltrombopag therapy and had platelet count at their baseline levels. Equal num-bers of megakaryocytic progenitors were divided between each channel for the ex vivo culture.

Patients with Inherited Thrombocytopenias have normal or slightly increased serum levels of TPO (*Zaninetti et al., 2020*), thus, in vivo hematopoietic progenitors are exposed to stimuli from endoge-nous TPO simultaneously with Eltrombopag treatment. To mimic this condition faithfully, all the sam-ples were cultured in the presence of 10 ng/mL recombinant human TPO alone or in combination with 500 ng/mL Eltrombopag (*Figure 4A*). Insights into the efficacy of Eltrombopag effects ex vivo were gained by simultaneously analyzing megakaryocyte differentiation at day 14 for each disorder. Specifically, cells were washed out of the device and analyzed. We observed comparable megakar-yocyte maturation in terms of cell size (*Figure 4B*), ploidy profile (*Figure 4C*), and expression of line-age-specific markers (*Figure 4D and E*), with and without Eltrombopag. However, the combination of TPO and Eltrombopag resulted in a significant two-fold increase in the output of mature megakar-yocytes with respect to TPO alone for both *ANKRD26*-RT and *MYH9*-RD patients (*Figure 4F*).

## Assessment of proplatelet formation to inform on mechanisms of action

Confocal microscopy analysis of 3D scaffolds revealed a homogeneous distribution of CD61$^+$ mega-karyocytes throughout the entire construct in both culture conditions, with more clusters in the pres-ence of Eltrombopag, from both *ANKRD26*-RT and *MYH9*-RD (*Figure 5Ai-iv*). Further, in the presence of Eltrombopag, megakaryocytes underwent characteristic cytoplasmic rearrangements typical of proplatelets (*Figure 5Aii,iv*). β1-tubulin staining of megakaryocytes harvested from the

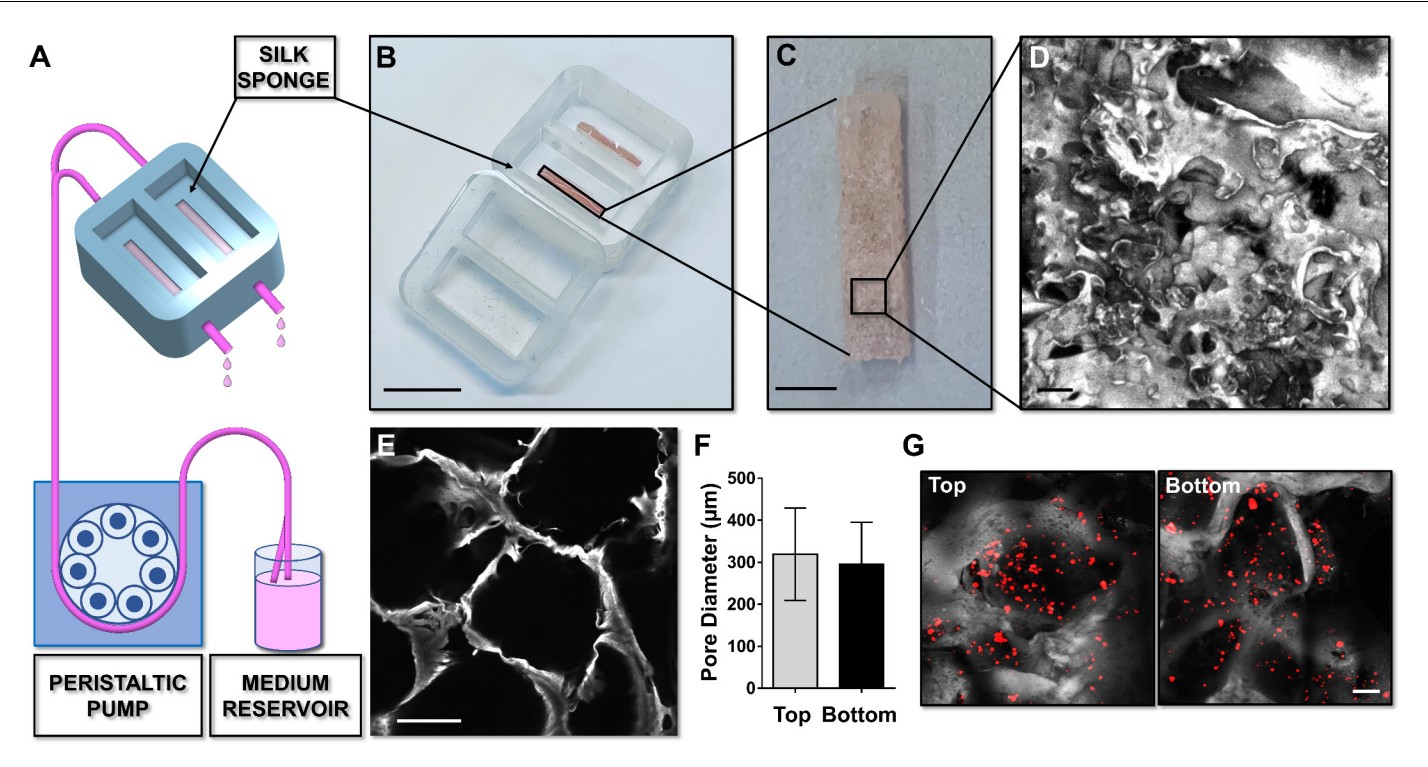

**Figure 2.** Silk sponge bone marrow perfusion system. (**A–C**) A peristaltic pump drives perfusion of the cell culture medium from a reservoir to the device equipped with a silk fibroin sponge prepared directly inside the chamber by dispensing an aqueous silk solution mixed with salt particles (scale bar B = 1.5 cm; scale bar C = 2 mm). After leaching out the salt, the resulting porous silk sponge can be sterilized. (**D,E**) Confocal microscopy reconstruction of the silk sponge showed the presence of an interconnected alveolar network (scale bar D = 200 µm; scale bar E = 150 µm). (**F**) The analysis of pore diameters measured on the top and bottom of the scaffold demonstrated no significant differences throughout the scaffold. Results are presented as mean ± SD (n = 150 pore/condition, p=NS). (**G**) Confocal microscopy analysis of CFSE[+] cells cultured within the silk scaffold (red = CFSE; gray = silk; scale bar = 50 µm). The full data set is provided in *Figure 2—source data 1*.

The online version of this article includes the following source data for figure 2:

**Source data 1.** Analysis of the pore diameter of the silk scaffolds.

device and seeded onto fibronectin-coated coverslips consistently highlighted that TPO in combination with Eltrombopag supported the extension of multiple branched shafts resembling nascent platelets at their terminal ends (*Figure 5Av-viii*) and a significant increase in the percentage of proplatelet-forming megakaryocytes in both *ANKRD26*-RT (TPO: 3 ± 2.6%; TPO plus Eltrombopag: 7.7 ± 4.4%) and *MYH9*-RD (TPO: 1.5 ± 1%; TPO plus Eltrombopag: 4.4 ± 4.3%) (*Figure 5B*).

## Ex vivo platelet count as a predictor of drug efficacy

Since the desired ultimate effect of Eltrombopag in patients with *ANKRD26*-RT or *MYH9*-RD is an increase in platelet count, platelet production was the most pertinent parameter evaluated in our ex vivo model. First we tested the possibility to harvest and count ex vivo produced platelets by perfusing the scaffolds cultured with megakaryocytes from healthy controls in the presence of TPO alone or TPO in combination with Eltrombopag. On day 15 of culture, each channel of the device was connected to a peristaltic pump at the inlet and a gas-permeable collection bag at the outlet. The number of ex vivo platelets produced was assessed and counted with a bead standard by flow cytometry after 4 hr of perfusion, at 37°C and 5% $CO_2$ (*Figure 6A*). The mean absolute number of collected platelets was 24 × 10$^4$/scaffold (range 18−35 × 10$^4$) in the presence of TPO, with a significant 1.7-fold increase in the presence of TPO in combination with Eltrombopag (p<0.05).

To test whether our device could predict the patient-specific response to Eltrombopag, we performed a systematic study comparing the extent of platelet production ex vivo to the platelet response observed in vivo in the same patients (*Zaninetti et al., 2019*; *Zaninetti et al., 2020*).

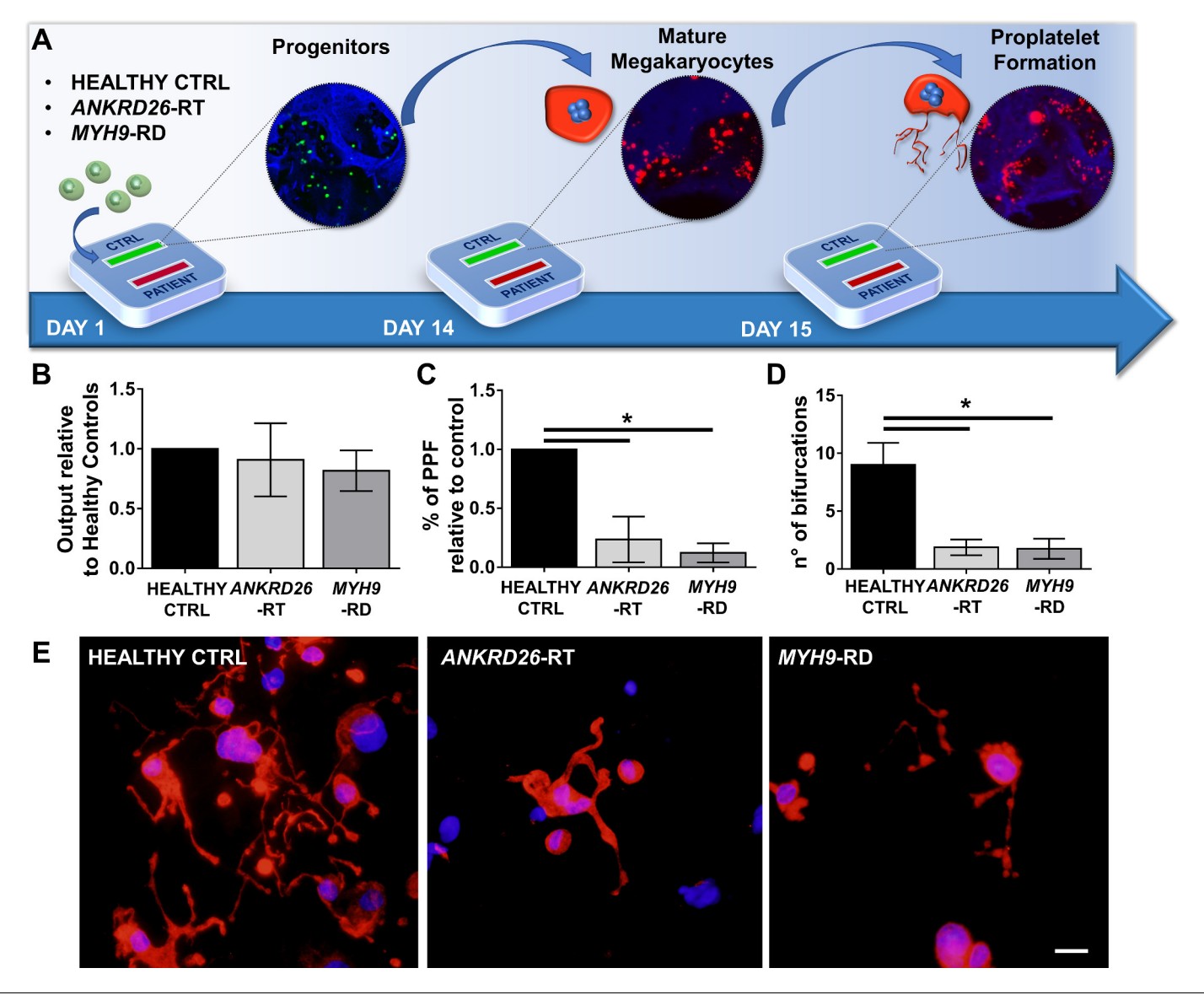

**Figure 3.** Modeling physiological and pathological megakaryopoiesis. (A) Megakaryocytes were differentiated from healthy controls and patients affected by *MYH9*-RD and *ANKRD26*-RT patients and cultured into the bone marrow device in presence of 10 ng/mL TPO. (B) Output of CD41⁺CD42b⁺ megakaryocyte at the end of differentiation relative to healthy controls (n = 12 Healthy Controls, n = 12 *MYH9*-RD; n = 12 *ANKRD26*-RT) (C) Percentage of proplatelet formation relative to healthy controls (n = 12 Healthy Controls, n = 12 *MYH9*-RD; n = 12 *ANKRD26*-RT; *p<0.01). (D) The number of proplatelet bifurcation per single megakaryocytes in healthy controls and patients (n = 12 Healthy Controls, n = 12 *MYH9*-RD; n = 12 *ANKRD26*-RT; *p<0.01). (E) Representative immunofluorescence staining of proplatelet structure (red=β1-tubulin; blue = nuclei; scale bar = 20 µm). All results are presented as mean ± SD. Data from the treatment of healthy controls in the presence of TPO and TPO +EPAG are shown in *Figure 3—figure supplement 1*. The full data set is provided in *Figure 3—source data 1*.

The online version of this article includes the following source data and figure supplement(s) for figure 3:

**Source data 1.** Analysis of megakaryocyte differentiation and proplatelet formation in healthy controls and patients.
**Figure supplement 1.** Eltrombopag is effective in promoting thrombopoiesis from healthy controls.
**Figure supplement 1—source data 1.** Analysis of megakaryocyte output and proplatelet formation in healthy controls in the presence of EPAG.

Samples were perfused in the same standardized conditions used with healthy controls. After perfusion, ex vivo collected platelets exhibited the β1-tubulin coil at their periphery, typically present in peripheral blood platelets (*Figure 6B*), further supporting the physiological relevance of the reproduced bone marrow environment for replicating in vivo thrombopoiesis. Ex vivo collected platelets

**Table 1.** Main features of the study population.

|  | *ANKRD26*-RT | *MYH9*-RD |
|---|---|---|
| Total samples, no. | 12 | 12 |
| M/F | 9/3 | 5/7 |
| Age - mean [range], years | 46 [22-67] | 48 [26-59] |
| Platelet count - mean [range] x10$^9$/L | 32 [9-75] | 29 [5-69] |

were double-stained with anti-CD41 and anti-CD42b antibodies and counted by flow cytometry (*Figure 6C*). The number of CD41$^+$CD42b$^+$ platelets collected per single channel increased significantly when treated with TPO in combination with Eltrombopag with respect to TPO alone, in both *ANKRD26*-RT and *MYH9*-RD groups (*Table 2*). However, while all samples from healthy controls responded to the treatment with Eltrombopag, in patients the platelet response was variable, with some samples demonstrating a slight or no increase in ex vivo platelet production. The same variability was present during the treatment in vivo (*Zaninetti et al., 2019*; *Zaninetti et al., 2020*). When the increase in platelet count obtained ex vivo in response to Eltrombopag was compared with the increase in platelet count observed in vivo after Eltrombopag administration in the same patients (*Table 2*), there was a statistically significant correlation (R square = 0.78; p<0.0001) (*Figure 6D*). The scientific relevance of this correlation was supported by evidence that the interpolation of the platelet count obtained in vivo after Eltrombopag administration with the megakaryocyte output calculated ex vivo did not register the same correlation (R square = 0.35) (*Figure 6E*), suggesting that ex vivo platelet count is the candidate parameter that is likely to predict the patients' response better.

## Device validation with induced pluripotent stem cells from patients

Patient heterogeneity at the genetic and phenotypic levels is an increasingly important consideration in understanding the evolution of disease and resistance to treatment. Induced pluripotent stem cells (iPSCs) represent a useful tool to study disease mechanisms and testing drugs. Thus, iPSC clones were generated from one *MYH9*-RD patient and one healthy control. Clones were tested for pluripotency and found positive for OCT4, NANOG, and SOX2 by qRT-PCR analysis (*Figure 7—figure supplement 1A*). SOX2, OCT4, NANOG, TRA 1–81, and SSEA4 marker expression was also confirmed by immunofluorescence analysis (*Figure 7—figure supplement 1B*). Both control and patient clones displayed a normal diploid karyotype (control: 46,XX; patient: 46,XY) without noticeable abnormalities (*Figure 7—figure supplement 2*).

Megakaryocyte differentiation of iPSC clones was confirmed in liquid culture conditions (*Figure 7A* and *Figure 7—figure supplement 3*) and demonstrated that *MYH9*-RD iPSCs present a defect in proplatelet formation (control: 5.6 ± 1.6%; *MYH9*-RD: 3.0 ± 1.1%) and branching (n° of bifurcations: control: 6.3 ± 2.8; *MYH9*-RD: 1.5 ± 1) with respect to control iPSCs (*Figure 7B–D*).

To validate their use within the bone marrow tissue bioreactor, megakaryocyte progenitors from iPSC clones were sorted at day 14 of differentiation based on the expression of CD61$^+$, an early progenitor lineage-specific marker, and cultured for an additional 5 days within the device in the presence of 50 ng/mL TPO supplemented or not with 500 ng/mL Eltrombopag (*Figure 8A*). The disease clones showed comparable megakaryocyte maturation in terms of cell size (*Figure 8Bi–ii*) and expression of CD41 and CD42b (*Figure 8C and D*) in the presence of TPO or TPO plus Eltrombopag. 3D reconstruction of cell cultures from different *MYH9*-mutated clones revealed an increased number of megakaryocytes throughout the scaffold in the presence of TPO plus Eltrombopag (*Figure 8Biii-iv*), paralleled by an increased percentage of proplatelets (*Figure 8Biii-iv*) having more branches with respect to TPO alone (*Figure 8Bv-viii*). Statistical analysis demonstrated a significant increase in cell proliferation in the presence of TPO plus Eltrombopag with respect to TPO alone (*Figure 8E*). Further, after perfusion, significantly increased platelet count was observed under treatment with Eltrombopag (*Figure 8F*).

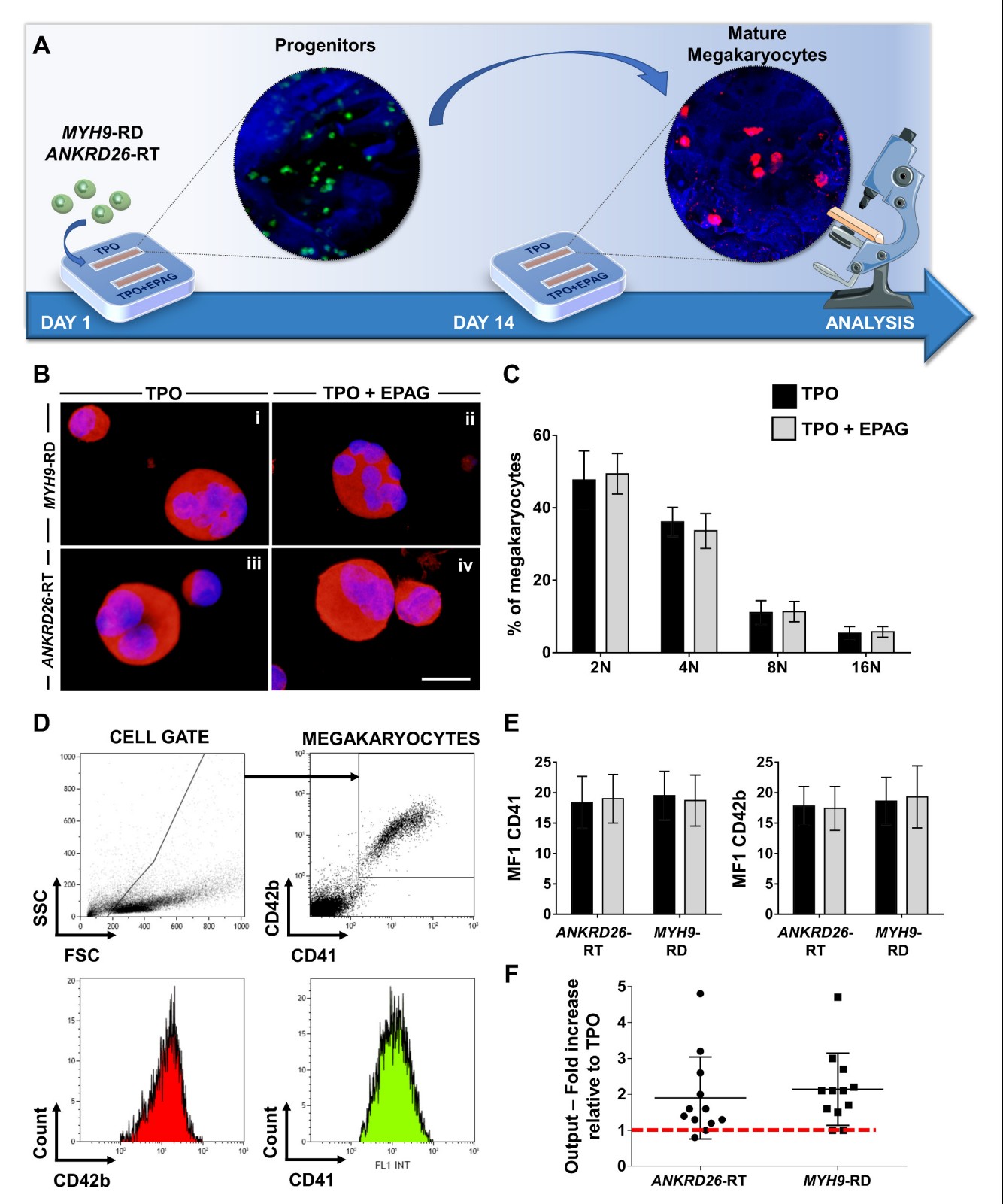

**Figure 4.** Eltrombopag promotes megakaryocyte differentiation ex vivo. (**A**) Megakaryocytes were differentiated from peripheral blood progenitors of patients affected by *MYH9*-RD or *ANKRD26*-RT and cultured in the silk bone marrow tissue device in the presence of 10 ng/mL TPO supplemented or not with 500 ng/mL Eltrombopag (EPAG) and analyzed. The figure of the microscope was adapted from Servier Medical Art licensed under a Creative Commons Attribution 3.0 Unported License (https://smart.servier.com). (**B**) Representative immunofluorescence staining of CD61 (red = CD61;
*Figure 4 continued on next page*

*Figure 4 continued*

blue = nuclei; scale bar = 25 μm) and (**C**) analysis of ploidy levels at the end of the culture (TPO: n = 3 *MYH9*-RD; n = 3 *ANKRD26*-RT; TPO +EPAG: n = 3 *MYH9*-RD; n = 3 *ANKRD26*-RT; p=NS). (**D**) Representative flow cytometry analysis of CD41$^+$CD42b$^+$ megakaryocytes at the end of the culture and (**E**) statistical analysis of mean fluorescence intensity (MFI) of the markers (TPO: n = 12 *MYH9*-RD; n = 12 *ANKRD26*-RT; TPO +EPAG: n = 12 *MYH9*-RD; n = 12 *ANKRD26*-RT; p=NS). (**F**) Output was calculated as the fold increase in the percentage of CD41$^+$CD42b$^+$ cells in presence of TPO +EPAG with respect to the percentage of double-positive cells in presence of TPO alone (*ANKRD26*-RT: n = 12, p<0.05; *MYH9*-RD: n = 12, p<0.01). All results are presented as mean ± SD. The full data set is provided in *Figure 4—source data 1*.

The online version of this article includes the following source data for figure 4:

**Source data 1.** Analysis of megakaryocyte differentiation.

## Discussion

Allogeneic platelet transfusions are widely used to treat acute bleeding in patients with thrombocytopenia of any origin and are also used to prevent bleeding in subjects who developed short-lasting, severe thrombocytopenia after chemotherapy or in those patients with more chronic thrombocytopenia in need of a procedure. However, platelet concentrates are not indicated for the prevention of hemorrhages in chronically thrombocytopenic patients for many reasons: they lose efficacy due to alloimmunization, acute reactions may occur, and transmission of infectious diseases is possible. Thus, platelet transfusions are not chronically administered to patients with Inherited Thrombocytopenia unless their platelet count is extremely low, and their risk of bleeding is relatively high. There is a need for alternative agents or approaches that could increase platelet count in these and chronic thrombocytopenic conditions.

TPO-receptor agonists stimulate megakaryopoiesis and platelet production. Eltrombopag and/or Romiplostim and/or Avatrombopag are currently approved for the treatment of primary immune thrombocytopenia at various stages of ITP in adults and children (*Bussel, 2009*), thrombocytopenia related to liver disease if a procedure is needed, and severe acquired aplastic anemia (*Olnes et al., 2012*). Small clinical trials and case reports have suggested that TPO receptor agonists are also effective in increasing platelet counts in patients with certain forms of Inherited Thrombocytopenia (*Rodeghiero et al., 2018*) and that at least Eltrombopag could be used to replace platelet transfusions to prepare patients to undergo hemostatic challenges (*Zaninetti et al., 2019*). Indeed, a few patients have successfully received long-term treatment with TPO-receptor agonists, potentially paving the way for chronic treatment of these previously untreated forms of thrombocytopenia. However, platelet response to these drugs was variable among different patients, and sometimes the drugs were ineffective (*Gerrits et al., 2015*; *Zaninetti et al., 2019*; *Zaninetti et al., 2020*).

Here, we have developed a miniaturized 3D bone marrow tissue model that ex vivo reproduces in vivo platelet biogenesis in such a way that it allows us to predict response to drugs on a single patient basis. The major advantages of this device over the existing bone marrow models include rapid customization and manufacturing, handling ease, and the implementation of small silk-based three-dimensional scaffolds to allow the seeding of small amounts of adult megakaryocyte progenitors that are cultured for several days, perfused in parallel and simultaneously, to compare platelet production under different treatments (*Table 3*). As proof of principle, we applied our system to study thrombocytopenic patients affected by *ANKRD26*-RT and *MYH9*-RD who were treated with the TPO-receptor agonist Eltrombopag (*Zaninetti et al., 2020*). According to our clinical data, the extent of platelet response to Eltrombopag in patients with *ANKRD26*-RT was lower than that in *MYH9*-RD (*Zaninetti et al., 2019*; *Zaninetti et al., 2020*). This range of variability was reflected ex vivo, clearly demonstrating that our tissue model can efficiently predict both the positive and negative response to Eltrombopag in individual patients and allow more personalized treatment, reducing the number of non-responders unnecessarily exposed to potential side effects of the treatment and ineffective preparation for procedures. In the future, patients might be able to create their own platelets and thus avoid most if not all of the complications discussed above. The study has several limitations. First, only patients affected by *ANKRD26*-RT and *MYH9*-RD were investigated; however, they are among the most frequent forms of Inherited Thrombocytopenia worldwide. For the many other forms of Inherited Thrombocytopenia, we do not know if our ex vivo predictive system will be similarly predictive. Second, since in vivo clinical trials for patients with Inherited Thrombocytopenia thus far have been limited to Eltrombopag; we, therefore, chose not to include either Romiplostim

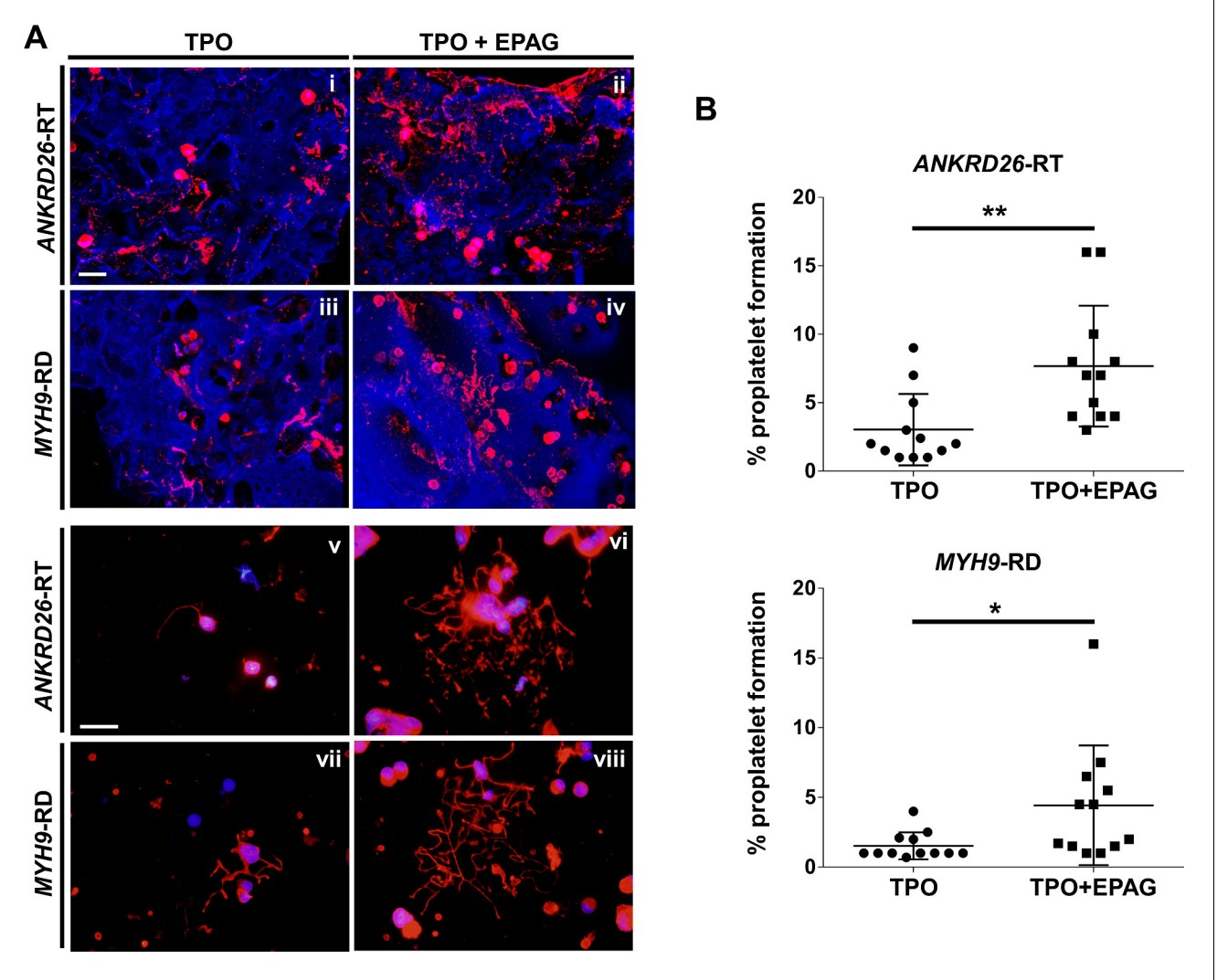

**Figure 5.** Eltrombopag sustains increased proplatelet formation ex vivo. (**A**) Confocal microscopy analysis of 3D megakaryocyte culture imaged at the end of differentiation. Megakaryocytes were elongating proplatelet shafts, which assembled nascent platelets at their terminal ends, within the hollow space of silk pores (red = CD61, blue = silk) (scale bars = 50 μm). (**A**v-viii) Analysis of proplatelet structure was performed by immunofluorescence staining of the megakaryocyte-specific cytoskeleton component β1-tubulin (red=β1-tubulin; blue = nuclei; scale bar = 25 μm). In both diseases, the representative pictures show increased elongation and branching of proplatelet shafts in presence of TPO +EPAG with respect to TPO alone. (**B**) The percentage of proplatelet forming megakaryocytes was calculated as the number of cells displaying long filamentous pseudopods with respect to the total number of round megakaryocytes per analyzed field (TPO: n = 12 *MYH9*-RD; n = 12 *ANKRD26*-RT; TPO +EPAG: n = 12 *MYH9*-RD; n = 12 *ANKRD26*-RT; **p<0.01; *p<0.05). All results are presented as mean ± SD. The full data set is provided in *Figure 5—source data 1*.
The online version of this article includes the following source data for figure 5:

**Source data 1.** Analysis of proplatelet formation.

or Avatrombopag (*Bussel, 2018*; *Ghanima et al., 2019*) in our ex vivo testing. Third, the model was not tested to assess the effect of drugs that negatively impact platelet production, such as chemotherapy. Nonetheless, these limitations can readily be overcome by an additional study of the various permutations discussed. Viewed from this perspective, the studies reported here provide motivation and rationale for extending the model to allow identification of the impact of various molecules on platelet production for each patient. Furthermore, this ex vivo approach may be useful to study drugs not only in diseases characterized by thrombocytopenia but also in those with thrombocytosis.

Inherited Thrombocytopenias each represent a prototype of thrombocytopenias deriving from defective platelet biogenesis within the bone marrow. For many Inherited Thrombocytopenias, the mechanisms of defective platelet production remain unknown. Understanding the cause of

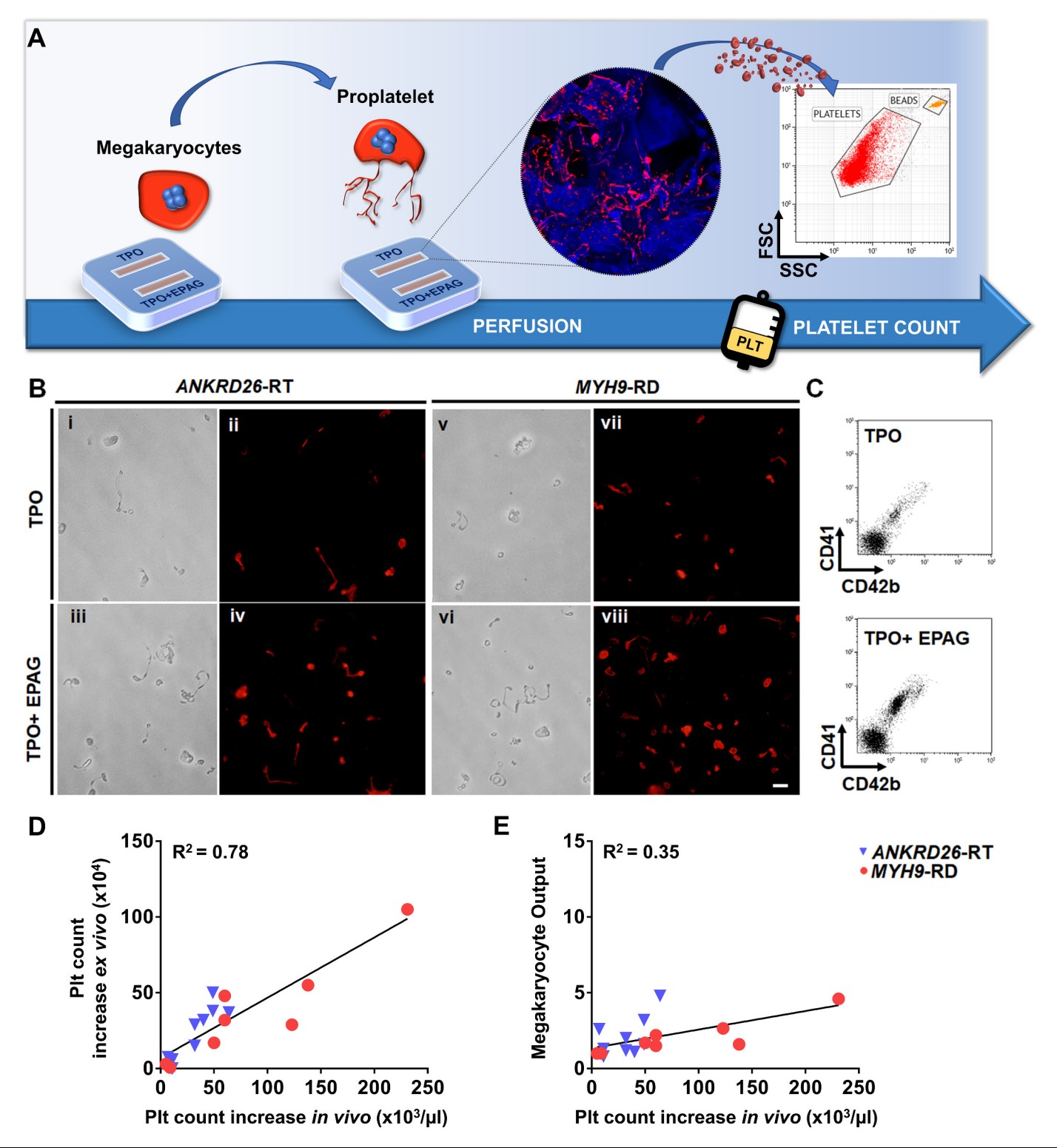

**Figure 6.** Ex vivo platelet count for predicting response to treatments. (**A**) The flow chamber was perfused with culture media and released platelets collected into gas-permeable bags before counting by flow cytometry. (**B**) Light microscopy and immunofluorescent analysis of the collected medium demonstrated the presence of large pre-platelets, dumbbells, and little discoid platelets having the microtubule coil typically present in resting platelets (red=β1-tubulin, scale bars = 10 µm). (**C**) Representative flow cytometry analysis of expression of CD41 and CD42b surface markers. (**D**) Analysis of the correlation between the increase of platelet count analyzed ex vivo and the increase of platelet count observed in vivo from the same patients. For the ex vivo analysis, platelet count was calculated by flow cytometry with counting beads (n = 8 *MYH9*-RD; n = 9 *ANKRD26*-RT). (**E**)

*Figure 6 continued on next page*

*Figure 6 continued*

Analysis of the correlation between ex vivo megakaryocyte output and the increase of platelet count observed in vivo from the same patients. (n = 8 *MYH9*-RD; n = 9 *ANKRD26*-RT). The full data set is provided in *Figure 6—source data 1*.

The online version of this article includes the following source data for figure 6:

**Source data 1.** Analysis of platelet count and megakaryocyte output ex vivo, and correlation with platelet count in vivo.

thrombocytopenia in these diseases could define the most suitable treatment for each disorder and identify both novel potential targets and either novel drugs or novel uses of existing drugs. Current 2D assays for functional assessment of megakaryocytes do not effectively monitor the final stage of maturation, in particular proplatelet spreading, platelet formation, and platelet release (*Balduini et al., 2016*). By recreating megakaryocyte maturation from stem cells to platelet release, our miniaturized 3D bone marrow model demonstrated the ability to reproduce these key steps of thrombopoiesis, including alterations observed in diseased states.

Megakaryocytes from patient-derived iPSCs reproduce the genetic background of peripheral-blood derived megakaryocytes and, thus, can be used to systematically study disease mechanisms and test candidate drugs. iPSCs represent a potentially unlimited source of megakaryocytes that can be frozen and made available on-demand without having to rely on frequent collection of patients' blood. We hypothesized that combining the 3D bone marrow tissue model and iPSC technologies would be instrumental in addressing critical clinical needs for a more specific understanding of the mechanism of action of TPO-receptor agonists in patients. The possibility of generating iPSC clones from fibroblasts of a *MYH9*-RD patient has been previously shown (*Tangprasittipap et al., 2019*), though, their differentiation potential was not proven. We derived iPSCs from hematopoietic stem and progenitor cells of one *MYH9*-RD patient and analyzed separately three different clones. The breakthrough of our approach includes an in-depth quality check of the iPSC clones to minimize heterogeneity and maximize replicability over-time. No defects in megakaryocyte maturation were observed from all the clones, while proplatelet formation was decreased as compared to healthy control clones.

Besides its ability to stimulate megakaryopoiesis, Eltrombopag has also been well-demonstrated to promote multilineage hematopoiesis in patients with acquired bone marrow failure syndromes (*Olnes et al., 2012*). Although the exact mechanisms of its effects on hematopoietic progenitor cells are not completely clear, Kao et al. recently demonstrated a stimulatory effect on stem cell self-

**Table 2.** Main features of the study population treated with Eltrombopag in vivo and ex vivo.

| | *ANKRD26*-RT | *MYH9*-RD |
|---|---|---|
| Patients treated with Eltrombopag in vivo and ex vivo, no. | 6[†] | 7[*] |
| M/F | 6/3 | 2/6 |
| Age - mean [range], years | 47 [22-67] | 45 [31-59] |
| Platelet count at baseline IN VIVO mean [range], x10$^9$/L | 36 [12-75] | 24 [5-69] |
| Increase of platelet count after Eltrombopag treatment IN VIVO‡ - mean [range], x10$^9$/L | 34 [7-64] | 88 [5-231] |
| Platelet count EX VIVO – TPO mean [range], x10$^4$ | 8.3 [6-13] | 7.8 [5-12] |
| Increase of platelet count EX VIVO§ – TPO + EPAG mean [range], x10$^4$ | 24 [0–50] | 36 [1-105] |

[*] of whom one patient repeated two times ex vivo.

[†] of whom three patients repeated two times ex vivo.

[‡] Increase of platelet count with Eltrombopag with respect to baseline.

[§] Increase of platelet count with Eltrombopag with respect to the untreated counterpart (TPO only).

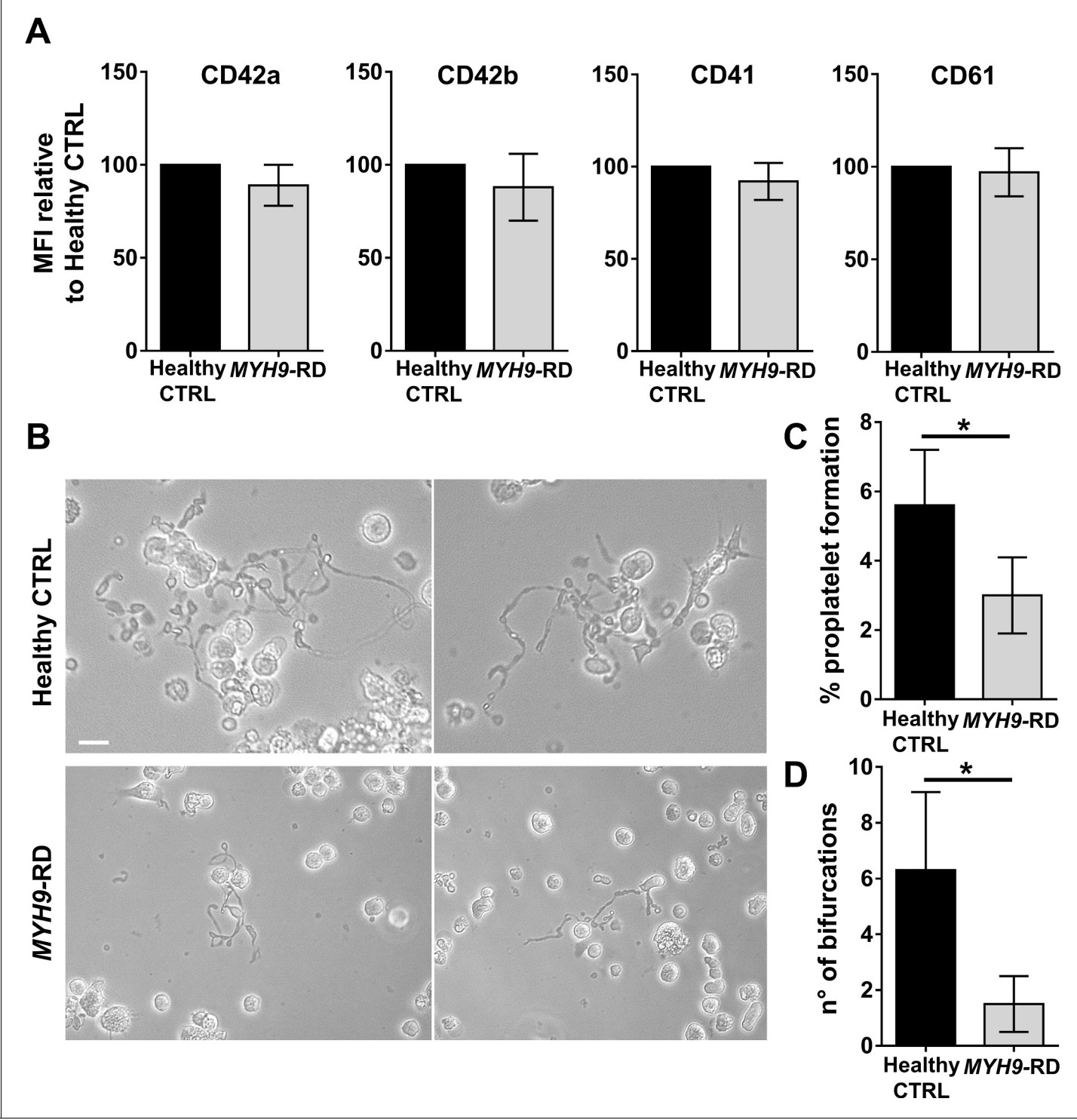

**Figure 7.** Assessment of iPSC megakaryocyte differentiation. (A) iPSC clones were cultured for 18 days and analyzed by flow cytometry to assess megakaryocytes differentiation. Histograms show the mean fluorescence intensity (MFI) of *MYH9*-RD clones for CD42a, CD42b, CD41, and CD61 markers, relative to healthy controls (n = 3 Healthy Controls, n = 3 *MYH9*-RD; p=NS). (B) Representative images of proplatelet forming-megakaryocytes at day 19 of culture from different iPSC clones. (C) Percentage of proplatelet formation from the different genotypes (n = 6 Healthy Control; n = 6 *MYH9*-RD (each clone repeated two times); *p<0.05. (D) The number of bifurcation per single megakaryocyte from the different genotypes (*p<0.01). All results are presented as mean ± SD. The full data set is provided in *Figure 7—source data 1*. Assessment of iPSC clone pluripotency is shown in *Figure 7—figure supplement 1*. Morphological and genomic characterization of iPSC clones is shown in *Figure 7—figure supplement 2*. iPSCs

*Figure 7 continued on next page*

*Figure 7 continued*

haematopoietic differentiation is shown in *Figure 7—figure supplement 3*. Embryoid Bodies and trilineage differentiation of iPSC clones is shown in *Figure 7—figure supplement 4*.

The online version of this article includes the following source data and figure supplement(s) for figure 7:

**Source data 1.** Analysis of iPSC differentiation and proplatelet formation.
**Figure supplement 1.** Characterization of iPSCs.
**Figure supplement 1—source data 1.** Analysis of gene expression in iPSCs.
**Figure supplement 2.** Morphological and genomic characterization of iPSC clones.
**Figure supplement 3.** iPSCs hematopoietic differentiation.
**Figure supplement 4.** Embryoid Bodies and trilineage differentiation of iPSC clones.

renewal independently of the TPO receptor-mediated through iron chelation-dependent molecular reprogramming (*Kao et al., 2018*). Our benchmark tests highlight that, besides platelet release, the 3D tissue model allowed us to track the effect of Eltrombopag on both progenitor cell and megakaryocyte functions, promising to provide a more comprehensive approach to study the effect of TPO-receptor agonists on hematopoietic stem and progenitor cells.

In summary, we developed a proof-of-concept system that in two weeks measures the impact of TPO-receptor agonists on megakaryopoiesis and platelet production of individual patients starting from a small amount of their peripheral blood (*Figure 9*). This silk-based technology, which can be produced and customized in 2 days, reaches the expectation of cost efficiency, time-saving, convenience, and personalization of modern therapeutic approaches. The data demonstrated that the ex vivo system could predict in vivo clinical response to Eltrombopag. The increase in the number of platelets collected in the ex vivo model was comparable to the increase of platelet count in vivo upon treatment with Eltrombopag. The broader impact of this work is in the design of tools to mimic the bone marrow ex vivo that can uncover mechanisms of impaired platelet production and enable testing of candidate drug treatments on platelet production using patient-derived cells. In the future, our system may serve as the foundation for highly integrated approaches to generate solutions for the ex vivo production of all blood cells for transfusion. The system may also represent a benchmark for pre-clinical testing of new therapeutic applications for Inherited Thrombocytopenias or other hematologic diseases, and for testing the effects of potentially toxic agents for the whole hematopoietic niche.

# Materials and methods

## Key resources table

| Reagent type (species) or resource | Designation | Source or reference | Identifiers | Additional information |
|---|---|---|---|---|
| Gene (human) | ANKRD26 – ankyrin repeat domain 26 | GenBank | THC2, bA145E8.1 | OMIM 188000 |
| Gene (human) | MYH9 – myosin heavy chain 9 | GenBank | BDPLT6, DFNA17, EPSTS, FTNS, MATINS, MHA, NMHC-II-A, NMMHC-IIA, NMMHCA | OMIM 155100 |
| Genetic reagent (human) | STR D21S11, D7S820, CSF1PO, TH01, D13S317, D16S539, vWA, TPOX, D5S818 Amelogenin | Promega GenePrint | Genetic multi-locus human DNA profile Genetic loci | Cell line authentication |
| Cell line (human) | Human embryonic stem cell | ISENET Biobank | Embryonic stem cell line RCe021-A (RC-17) Rosalin cells, RRID:CVCL_L206 | Control embryonic stem cell (human) |

*Continued on next page*

*Continued*

| Reagent type (species) or resource | Designation | Source or reference | Identifiers | Additional information |
|---|---|---|---|---|
| Cell line (human) | iPSC generated from healthy control | This paper | CTR2#6 | Control iPSC (human) |
| Cell line (human) | iPSC generated from healthy control | This paper | CPN | Control iPSC (human) |
| Cell line (human) | iPSC generated from *MYH9*-RD patient | This paper | MH | Patient iPSC (human) |
| Transfected construct (human) | Sindai virus OCT4, KLF4, Sox2, cMyc | Life Technologies | CytoTune-iPS Sendai Reprogramming | Generation of iPSCs (human) |
| Biological sample (human) | Peripheral Blood | - | - | - |
| Commercial assay or kit | CytoTune-iPS 2.0 Sendai Reprogramming Kit | Invitrogen/Thermo Fisher Scientific | #A16517 | Generation of iPSCs (human) |
| Commercial assay or kit | Venor GeM | Minerva Biolabs | #11–1250 | Mycoplasma detection |
| Commercial assay or kit | Qiagen DNA Mini Kit | Qiagen | #A31881 | DNA extraction |
| Commercial assay or kit | TruCount | Becton - Dickinson | #340334 | FACS Counting Beads |
| Commercial assay or kit | Slide-A-Lyzer Dyalisis Cassettes, 3.5K MWCO, 12 mL | Thermo Scientific | #66110 | Dyalisis |
| Commercial assay or kit | MiniMACS Starting Kit | Miltenyi Biotec | # 130-090-312 | Cell sorting |
| Antibody | Anti-Human CD61 (Mouse Monoclonal) | Beckman Coulter | #IM0540, RRID:AB_2889176 | 1:100 |
| Antibody | Anti - CD45 Microbeads (Mouse Monoclonal) | Miltenyi Biotec | #130-045-801, RRID:AB_2783001 | 20 µl / $10^7$ cells |
| Antibody | Anti - CD61 Microbeads (Mouse Monoclonal) | Miltenyi Biotec | #130-051-101 RRID:AB_2889174 | 20 µl / $10^7$ cells |
| Antibody | Alexa Fluor 594 (Goat Anti Mouse) | ThermoFisher Scientific | # A11005, RRID:AB_2534073 | 1:500 |
| Antibody | FITC anti-human CD41 Antibody (Mouse Monoclonal) | BioLegend | #303703, RRID:AB_314373 | 5 µl Megakaryocytic marker |
| Antibody | PE anti-human CD42b Antibody (Mouse Monoclonal) | BioLegend | #303905, RRID:AB_314385 | 5 µl Megakaryocytic marker |
| Antibody | a-FP (Mouse Monoclonal) | R and D system | #MAB1369, RRID:AB_2258005 | 1:50 Trilineage differentiation |
| Antibody | a-SMA (Mouse Monoclonal) | Millipore Sigma | #A2547, RRID:AB_476701 | 1:200 Trilineage differentiation |
| Antibody | bIII-Tubulin (Mouse Monoclonal) | Millipore Sigma | #MAB1637, RRID:AB_2210524 | 1:100 Trilineage differentiation |
| Antibody | SOX2 (Rabbit Monoclonal) | Abcam, | #Ab97959, RRID:AB_2341193 | 1:300 Pluripotency markers |

*Continued*

| Reagent type (species) or resource | Designation | Source or reference | Identifiers | Additional information |
|---|---|---|---|---|
| Antibody | Nanog (Mouse Monoclonal) | Millipore | #MABD24, RRID:AB_11203826 | 1:500 Pluripotency markers |
| Antibody | SSEA4 (Mouse Monoclonal) | Millipore | #MAB4304, RRID:AB_177629 | 1:50 Pluripotency markers |
| Antibody | Tra1-81 (Mouse Monoclonal) | Millipore | #MAB4381C3, RRID:AB_2889175 | 1:50 Pluripotency markers |
| Sequence-based reagent | GenePrint 10 System | Promega | #B9510 | STR genotyping |
| Peptide, recombinant protein | Recombinant Human Thrombopoietin (TPO) | Peprotech | #300–18 | Cytokine |
| Peptide, recombinant protein | Recombinant Human interleukin-11 (IL-11) | Peprotech | #200–11 | Cytokine |
| Peptide, recombinant protein | Recombinant Human interleukin-6 (IL-6) | Peprotech | #200–6 | Cytokine |
| Peptide, recombinant protein | Recombinant Human Fibroblast Growth Factor (FGF) | Peprotech | #100-18B | Cytokine |
| Peptide, recombinant protein | Recombinant Fms-related tyrosine kinase three ligand (Flt3L) | Peprotech | #300–19 | Cytokine |
| Peptide, recombinant protein | Recombinant Human Stem Cell Factor (SCF) | Peprotech | #300–07 | Cytokine |
| Peptide, recombinant protein | Recombinant Human VEGF$_{165}$ | Peprotech | #100–20 | Cytokine |
| Chemical compound, drug | CHIR 99021 | Tocris | #4423 | Pharmacological Inhibitor |
| Chemical compound, drug | Lithium Bromide (LiBr) | Sigma-Aldrich | #213225 | Silk processing |
| Chemical compound, drug | Penicillin – Streptomycin 100X | Euroclone | #EB3001D | Antibiotics |
| Chemical compound, drug | Paraformaldehyde (PFA) | Sigma - Aldrich | #158127 | Fixation |
| Chemical compound, drug | Triton X-100 | Sigma - Aldrich | #X100 | Permeabilization |
| Chemical compound, drug | Eltrombopag | Novartis | N/A | Drug Testing |
| Software algorithm | GeneMapper Software version 4.0 | Applied Biosystem | #4440915, RRID:SCR_014290 | Genotyping Analysis |

*Continued on next page*

*Continued*

| Reagent type (species) or resource | Designation | Source or reference | Identifiers | Additional information |
|---|---|---|---|---|
| Other | Human Fibronectin | Becton Dickinson | # 354008 | Silk functionalization |
| Other | Poly(lactic acid) | FormFutura | - | Scaffolding |
| Other | Pluronic F-127 | Sigma-Aldrich | #P2443 | 25% |
| Other | PDMS (SYLGARD 184 Silicone Elastomer) | Dow Corning | # 1673921 | Scaffolding |
| Other | Dulbecco's Phosphate Buffered Saline (PBS) | Euroclone | #ECB4053L | Saline buffer |
| Other | Essential 8 Flex media | Gibco/Thermo Fisher | #A2858501 | Culturing media |
| Other | StemSpan SFEM Medium | Voden | #09650 | Culture medium |
| Other | E8 medium | Gibco/Thermo Fisher Scientific | #A1517001 | Culture medium |
| Other | L-Glutamine 100X | Euroclone | #ECB300D | 1% |
| Other | Hoechst 33258 | Sigma - Aldrich | #861405 | 1:10000 |
| Other | ProLong Gold antifade reagent | Invitrogen | #P36980 | Microscopy |
| Other | CFSE Cell Division Tracker Kit | Biolegend | #423801 | 5 mM |
| Other | Geltrex | Gibco/Thermo Fisher Scientific | #A1413202 | Matrix |
| Other | VTN-N | Gibco/Thermo Fisher Scientific | #A14700 | Matrix |
| Other | KaryoMax Colcemid | Gibco - Sigma | 329749009 | Chromosome/metaphase banding Genomic stability |
| Other | Quinacrine | Sigma/Merck | Q3251 | Chromosome/metaphase banding Genomic stability |
| Other | aCGH | Agilent Technologies | SurePrint G3 Human CGH Microarray 8 × 60K | DNA Analytics software v. 5CNVs |

## Materials

*B. mori* silkworm cocoons were supplied by Tajima Shoji Co., Ltd. (Yokohama, Japan). Pharmed tubing was from Cole-Parmer (Vernon Hills, IL, USA). The immunomagnetic separation system was from Miltenyi Biotech (Bergisch Gladbach, Germany and Bologna, Italy). Recombinant human TPO, interleukin-6 (IL-6), interleukin-11 (IL-11), human bone morphogenetic protein 4 (BMP4), human vascular endothelial growth factor (VEGF), human fibroblast growth factor (FGF), human Fms-related tyrosine kinase three ligand (Flt3L), human stem cell factor (SCF) were from Peprotech (London, UK). CHiR 99021 was from TOCRIS. TruCount tubes and human fibronectin were from Becton Dickinson (S. Jose, CA, USA). The following antibodies were used: mouse monoclonal anti-CD61, clone SZ21, from Immunotech (Marseille, France); rabbit monoclonal anti-β1-tubulin was a kind gift of Prof. Joseph Italiano (Brigham and Women's Hospital, Boston, USA). Alexa Fluor conjugated secondary antibodies and Hoechst 33258 were from Life Technologies (Monza, Italy). Additional details can be found in the 'Key resources table'.

**Table 3.** Comparison of silk-bone marrow models for megakaryocyte culture.

|  | Di Buduo et al., 2015 | Di Buduo et al., 2017 | Current Manuscript |
|---|---|---|---|
| Size of the cell-seeding well | $15 \times 20 \times 5$ mm ($1500$ mm$^3$) | $3.5 \times 20 \times 5$ mm ($350$ mm$^3$) | $2 \times 15 \times 3.5$ mm ($105$ mm$^3$) |
| No. of chambers that can be perfused in parallel | Max. 2 | Max. 1 | >2 (up to at least 4) |
| Source of blood | Human Umbilical Cord Blood | Human Umbilical Cord Blood | Human Peripheral Blood |
| Type of hemopoietic stem and progenitor cells | $CD34^+$ | $CD34^+$ | $^-$ $CD45^+$ $^-$ $CD34^+$-derived iPSCs |
| Type of cells seeded | $CD34^+$-derived megakaryocytes | $CD34^+$-derived megakaryocytes | - $CD45^+$-derived megakaryocyte progenitors - iPSC-derived megakaryocyte progenitors |
| No. of cells seeded | $2.5 \times 10^5$ | $4 \times 10^5$ | $5 \times 10^4$ |
| Time of the culture | 24 hr | 24 hr | >7 days |
| Major application | Studying mechanisms of thrombopoiesis | Proof of concept for scaling up platelet production | Drug Testing in individual patients |

## Production of the drug-testing device

The chamber was manufactured using 3D FDM printing technology and a biocompatible silicon molding approach. The modeling of the bioreactor was created using CAD software (OnShape, Fusion360, or Inventor2017) and used to generate 3D negative mold components exported as STL (Standard Triangulation Language) files, sliced with Slic3R PE, and export to the FDM 3D printer Prusa i3 MK3S (Prusa Research, Czech Republic). The printing is done using a poly(lactic acid) (PLA) high-temperature filament of 1,75 mm (FormFutura, Netherland) deployed in layers of 100 µm by a 0.25 mm nozzle. After printing, the mold was cured in an oven at 100°C for 20 min to increase mechanical properties. To produce the perfusion channel, 21G needles were disposed in the dedicated holes and sealed with a gel of 25% Pluronic F-127. The molding was performed using a polydimethylsiloxane (PDMS) (Sylgard184, Dow Corning), mixed in a 10:1 ratio of base material and curing agent. The selected material is stable both at low and high temperatures (45–200°C) and it is resistant to UV, water, and solvents. The PDMS was poured into the 3D printed molds that were positioned into a vacuum chamber to remove all the air bubbles. The curing of the PDMS was performed in a dried oven at 70°C for 4 hr; the molds were then dissociated from the final silicon models sterilizable by autoclave. The chamber consisted of two wells of $10 \times 22$ mm, having a hollow cavity of $2 \times 15 \times 3.5$ mm enclosed in a block of $30 \times 30 \times 14$ mm and connected to the outside of the chamber through channels of 0.9 mm diameter. The luer adaptors for the inlet and outlet were mounted in the channel and sealed with biocompatible silicone adhesive MD7-4502 (Dow Corning, USA). Then, the modular flow chamber was equipped with a silk fibroin sponge functionalized with fibronectin as described previously (*Di Buduo et al., 2017*).

## Preparation of the silk fibroin solution

Silk fibroin aqueous solution was obtained from *B. mori* silkworm cocoons according to previously published literature (*Di Buduo et al., 2018*). Briefly, dewormed cocoons were boiled for 30 min in 0.02 M Na$_2$CO$_3$ solution at a weight to volume ratio of 10 g to 4 L. The fibers were rinsed for 20 min three times in ultrapure water and dried overnight. The dried fibers were solubilized for 4 hr at 60°C in 9.3 M LiBr at a weight to volume ratio of 3 g/12 mL. The solubilized silk solution was dialyzed against distilled water using a Slide-A-Lyzer cassette (Thermo Scientific, Waltham, MA, USA) with a 3500 MW cutoff for three days and changing the water a total of eight times. The silk solution was centrifuged at maximum speed for 15 min to remove large particulates and stored at 4°C. The concentration of the silk solution was determined by drying a known volume of the solution overnight at 60°C and massing the remaining solids.

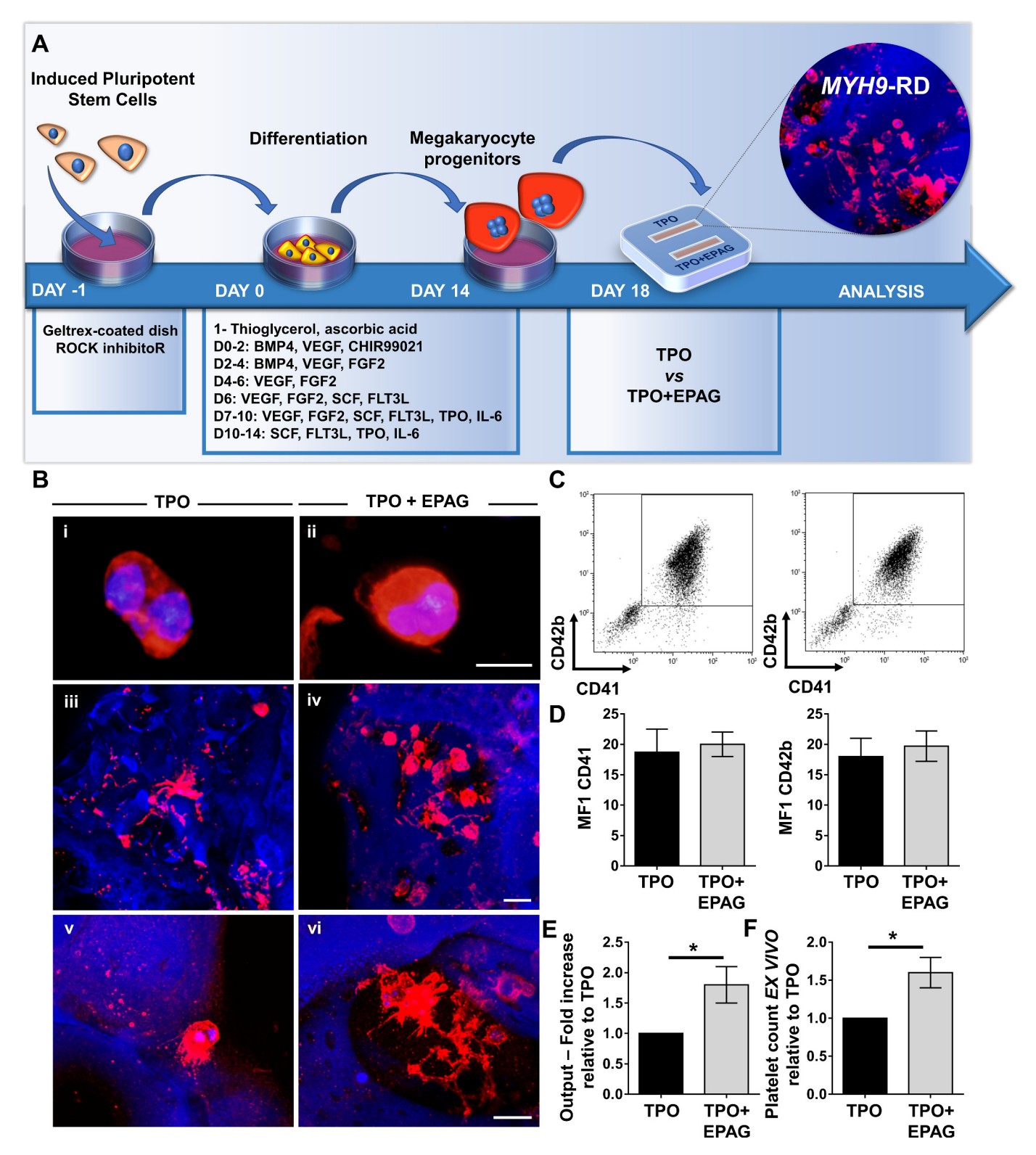

**Figure 8.** Validation of the system with iPSC mutated clones. (**A**) Megakaryocytes were differentiated from iPSCs of patients affected by *MYH9*-RD and cultured for 14 days in a petri dish before passing into the bone marrow device in presence of 10 ng/mL TPO supplemented or not with 500 ng/mL EPAG. (**B**) Representative immunofluorescence staining of CD61 megakaryocytes (**i,ii**) and confocal microscopy analysis (**iii-vi**) of 3D megakaryocyte culture imaged at the end of differentiation. Megakaryocytes are elongating proplatelet shafts, which assemble nascent platelets at their terminal ends,

*Figure 8 continued on next page*

 **eLife** Research article

Medicine

*Figure 8 continued*

within the hollow space of silk pores (red = CD61, blue = silk, scale bars = 50 µm). (C) Representative flow cytometry analysis of CD41$^+$CD42b$^+$ megakaryocytes at the end of the culture and (D) statistical analysis of mean fluorescence intensity (MFI) of the markers (n = 3 TPO, n = 3 TPO +EPAG; p=NS). (E) Output was calculated as the number of CD41$^+$CD42b$^+$ cells in presence of TPO +EPAG with respect to the percentage of double-positive cells in presence of TPO alone (n = 3 TPO, n = 3 TPO +EPAG; *p<0.05). (F) Platelet number was calculated by counting beads after perfusing the chamber. The fold increase was calculated as the number of ex vivo platelet count in the presence of TPO +EPAG with respect to TPO alone (n = 3 TPO, n = 3 TPO +EPAG; *p<0.05). All results are presented as mean ± SD. The full data set is provided in *Figure 8—source data 1*.

The online version of this article includes the following source data for figure 8:

**Source data 1.** Analysis of iPSC differentiation and platelet production in the presence of EPAG.

## Silk bone marrow fabrication and assembly

Silk solution (8% w/v) (*Lovett et al., 2007*) was mixed with 25 µg/mL fibronectin and dispensed into the modular chamber. NaCl particles (approximately 500 µm in diameter) were then sifted into the solution in a ratio of 1 mL to 2 g of NaCl particles. The scaffolds were then placed at room temperature for 48 hr and then soaked in distilled water for 48 hr to leach out the NaCl particles. The scaffolds were sterilized in 70% ethanol and finally rinsed five times in PBS for over 24 hr. Silk scaffolds were characterized by confocal, as subsequently described. Perfusion of the silk scaffold has been tested at different flow rates (5–50 µL/min) by using a peristaltic pump (ShenChen Flow Rates Peristaltic Pump - LabV1, China). The total volume collected after each test corresponded to that injected in the system by the pump.

## Patients

Human peripheral blood samples were obtained from healthy controls and thrombocytopenic patients after informed consent. All samples were processed following the ethical committee of the I.R.C.C.S. Policlinico San Matteo Foundation and the principles of the Helsinki Declaration. The main features of the 24 investigated samples from 20 different patients are reported in *Table 1*. For four patients, the analysis was performed on two different occasions, with very similar results. Diagnosis of *MYH9*-RD or *ANKRD26*-RT had been confirmed by genetic analysis in all the cases. All patients provided written informed consent for this study, which was approved by the Institutional Review Board of the IRCCS Policlinico San Matteo Foundation, Pavia, Italy. A sample of 15 mL of peripheral venous blood anticoagulated with ACD was collected for the analysis in the 3D bone marrow system. Thirteen patients had previously received a short-term course of Eltrombopag (3–6 weeks) either within a phase two clinical trial (*Zaninetti et al., 2020*) (n = 11) or in preparation for elective surgery (*Zaninetti et al., 2019*) (n = 2). In any case, Eltrombopag was given at the dose of 50 or 75 mg/day for 3 or 6 weeks (*Zaninetti et al., 2019*; *Zaninetti et al., 2020*). The in vivo clinical response to the drug was expressed as the absolute increase in platelet count at the end of Eltrombopag treatment with respect to baseline. Blood samples for this study were collected when patients were out of Eltrombopag therapy (minimum of 6 months to a maximum of 48 months washout). According to the fast pharmacokinetics of Eltrombopag (plasma elimination half-life approximately 21–32 hr), all patients had platelet count at their baseline levels at one month of follow-up after the discontinuation of the treatment (*Pecci et al., 2010*; *Zaninetti et al., 2020*). Thus, we do not expect that previous exposure to Eltrombopag may have influenced haematopoietic stem and progenitor cell functions after months.

## Human megakaryocyte differentiation within the silk bone marrow

CD45$^+$ hematopoietic progenitor cells from peripheral blood samples were separated by an immunomagnetic bead selection kit (Miltenyi Biotec, Bologna, Italy) and cultured for 6 days in a flask in presence in Stem Span media (StemCell Technologies, Canada) supplemented with 1% penicillin-streptomycin, 1% L-glutamine, 10 ng/mL TPO, IL-6, and IL-11 in the presence or not of 500 ng/mL Eltrombopag (Novartis) at 37°C in a 5% CO$_2$ fully humidified atmosphere, as previously described (*Bluteau et al., 2014*; *Pecci et al., 2009*).

On day 6, CD61$^+$ early megakaryocytic progenitors were sorted by immunomagnetic selection kit (Miltenyi Biotec, Bologna, Italy) and seeded for additional 8 days within the silk bone marrow model in presence of 10 ng/mL TPO supplemented or not with 500 ng/mL Eltrombopag.

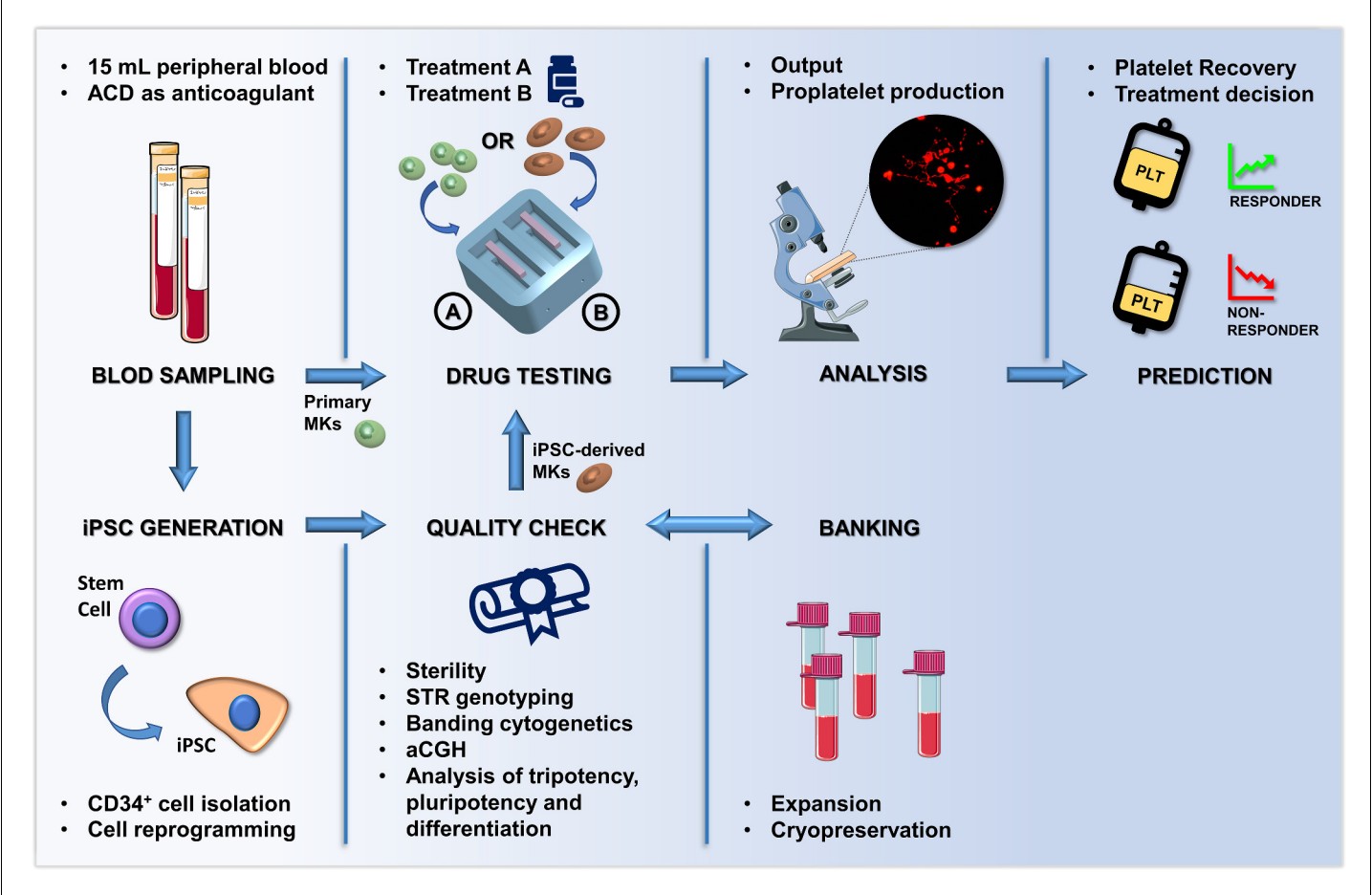

**Figure 9.** Summary of the proposed workflow. After sampling, hematopoietic stem and progenitors cell from patients can be differentiated into primary megakaryocytes (MKs) or transformed into induced Pluripotent Stem Cells (iPSCs). iPSC are subjected to quality check, expansion and banking. Megakaryocytic progenitors differentiated either from primary stem cells or iPSCs are seeded within a 3D bone marrow tissue device, cultured in the presence of the tested drug, and analyzed. After perfusion, platelets are collected and counted in order to assess patient-specific response. The figure of the microscope and tubes was adapted from Servier Medical Art licensed under a Creative Commons Attribution 3.0 Unported License (https://smart.servier.com).

On day 14 of differentiation, the chamber was sealed, and the outlet ports were connected to the outlet needles. Culture media-filled tubes were connected to the inlet needles. The chamber was placed into the incubator (37°C and 5% $CO_2$), and transfer bags for platelet collection were secured to the outlet ports. The peristaltic pump (ShenChen Flow Rates Peristaltic Pump - LabV1, China) was placed outside the incubator, and media was pumped for 4 hr at a flow rate of 10 µL/min, speed range: 0.18 rpm, perfusion pause: 120 s, perfusion run: 5 min, with a peristaltic pump.

Induced pluripotent stem cells generation: iPSCs were derived from one *MYH9*-RD patient with heterozygous g.103845T > A mutation and one healthy control providing their informed consent before the participation in this study and in accordance with the local ethical committee and the Declaration of Helsinki. At the time of the sampling, the patient was a 32-year-old male with a baseline platelet count of $50 \times 10^9$/L who had never been treated with Eltrombopag.

CD34$^+$ hematopoietic stem and progenitor cells were isolated from their peripheral blood using an immunomagnetic beads cell-sorting system (AutoMacs; Miltenyi Biotec, Paris, France) and amplified in serum-free media containing EPO (1 U/mL), FLT3L (10 ng/mL), G-CSF (20 ng/mL), IL-3 (10 ng/mL), IL-6 (10 ng/mL), SCF (25 ng/mL), TPO (10 ng/mL), and GM-CSF (10 ng/mL) for 6 days. Cells were then transduced with the CytoTune iPS 2.0 Sendai Reprogramming Kit (Thermo Fisher, Villebon-sur-Yvette, France) and the reprogramming was performed according to the manufacturer's instructions. Colonies with an ES-like morphology were manually isolated, expanded for a small

number of passages, and frozen. iPSCs were maintained on Essential eight or Essential 8 Flex media (Gibco/Thermo Fisher), on plates coated with N-truncated human recombinant vitronectin (Gibco). All derived clones were organized in colonies with defined edges and characterized by prominent nucleolus with a high nucleus-to-cytoplasm ratio (*Figure 7—figure supplement 2A*). Cell passages were performed using a solution of EDTA 0.5 mM in PBS 1X, or TrypLE 1X (Gibco).

## Conventional PCR mycoplasma detection

Supernatant samples from the generated iPSC lines were used for mycoplasma detection. A total of 100 μl supernatants were heated at 95°C for 10 min and centrifuged at 1000 g for 5 s to discard cellular debris. For the detection of mycoplasma species was used Venor GeM OneStep Mycoplasma Detection Kit for Conventional Polymerase chain reaction (PCR) (Minerva Biolabs) per manufacturer's instructions. PCRs were performed using a thermocycler. Amplified products were fractionated on 1.5% agarose and observed with Amersham Imager 680 (GE). All derived clones resulted negative for Mycoplasma screening.

## DNA isolation and quantification

Cell pellets from iPSC lines were used for DNA isolation. QIAamp DNA Mini Kit (Qiagen) was used following manufacturer instructions. DNA quantification was performed on Qubit. 2.0 Fluorometer (Thermo Fisher).

## STR genotyping

The genetic matching of the generated iPSC lines to the parental cells was confirmed by short tandem repeat (STR) analysis. The amplification was performed by PCR using ~1 ng/sample: nine autosomal STR molecular markers (D21S11, D7S820, CSF1PO, TH01, D13S317, D16S539, vWA, TPOX, D5S818) along with the gender determining marker Amelogenin with Promega GenePrint 10 Kit following manufacturer's recommended protocol. Appropriate positive and negative amplification controls were used as kit recommended guidelines. The amplified products were electrophoresed on an ABI Prism 3730xl Genetic Analyzer using an Internal Lane Standard 600 (Promega). Data generated were analyzed using GeneMapper Software version 4.0 (Applied Biosystems) following the manufacturer's instructions. Results demonstrated that iPSC clones and the blood donor perfectly matched the 10 loci tested (*Table 4*).

## Banding cytogenetics

Metaphase-chromosome spreads were prepared from 80% confluent cultures according to standard procedures. Actively dividing cells were treated with 10 ng/ml colcemid (Gibco KaryoMAX Colcemid solution in PBS, Thermo Fisher Scientific) for 16 hr (overnight) at 37°C. Cells were combined in 0,56% KCl for 20 min at 37°C and were fixed with methanol/acetic acid (3:1 v/v). Chromosome analysis was carried out by applying Q-banding by fluorescence using quinacrine (QFQ), according to routine procedures, following the guidelines of the International System for Chromosome Nomenclature 2009 (ISCN 2009) (*Shaffer et al., 2009*). Microscope observation was performed using the Fluorescence microscope Olympus BX63, fully equipped with quinacrine mustard filter and CCD camera and the acquisition and analysis of 'GenASIs' Software, version 8.1.0.47741 (Applied Spectral Imaging). On average, 25 metaphases were evaluated. No gross chromosomal alterations were observed by Q-banding (*Figure 7—figure supplement 2B*).

**Table 4.** STR-genotype of iPSCs and donor cells.
Numbers in each locus refer to the number of repeats in each allele.

| Cell line | TH01 | D21S11 | D5S818 | D13S317 | D7S820 | D16S539 | CSF1PO | AMEL | vWA | TPOX |
|-----------|------|--------|--------|---------|--------|---------|--------|------|-----|------|
| CPN | 9.3, 9.3 | 28, 33.2 | 12, 12 | 8,11 | 10,11 | 11,11 | 10,12 | X,X | 16,16 | 8,8 |
| MH | 7,7 | 32.2, 33.2 | 12,13 | 11,13 | 9,11 | 12,12 | 11,11 | X,Y | 15,16 | 11,11 |
| MH-1 | 7,7 | 32.2, 33.2 | 12,13 | 11,13 | 9,11 | 12,12 | 11,11 | X,Y | 15,16 | 11,11 |
| MH-11 | 7,7 | 32.2, 33.2 | 12,13 | 11,13 | 9,11 | 12,12 | 11,11 | X,Y | 15,16 | 11,11 |

## Array CGH analysis

Comparative genomic hybridization array (aCGH) analysis was performed using Agilent Human Genome CGH Microarray 60K kit (Agilent Technologies, Palo Alto, CA, USA), following the manufacturer's instructions. A sex-matched commercial DNA sample (Male, Promega, Milan, Italy) was used as reference DNA. Hybridization signals were analyzed using Feature Extraction software (v10.7) and DNA Analytics software (v5.0, Agilent Technologies, Palo Alto, CA, USA). Aberration Detection Method 2 (ADM2) algorithm (threshold 5.0) was used to identify DNA copy number aberrations. We applied a filtering option of a minimum of three aberrant consecutive probes (*Wu et al., 2007*) and a minimum absolute average log 2 ratio of 0.30. University of California Santa Cruz (UCSC) human genome assembly hg18 was used as a reference and copy number variations (CNVs) were identified with a database integrated into the Agilent Genomic Workbench analytic software. Log two ratios lower than −0.30 were classified as losses, those greater than 0.30 as gains. The analysis revealed CNVs for the one iPSC clone, consisting of amplification of 36 kb and 175 Kb, on chromosomes 2 and 3, respectively, involving genes with no significant associated clinical phenotype. The clone from healthy control did not reveal any CNVs (*Table 5*).

## RNA isolation and cDNA synthesis

Total RNA from each cell line was isolated using TRIzol reagent (Sigma) following the manufacture's protocol. For semi-quantitative PCR (qPCR) experiments, equal amounts of total RNA (1.3 µg) were reverse transcribed by using the RevertAid First Strand cDNA Synthesis Kit (Thermoscientific). The complementary DNA (cDNA) samples were used for validation of the self-renewal stem cell markers using RT-PCR analysis. qRT-PCR was assessed in triplicate on at least two independent biological replicates by the DDCt method on Rotor-Gene Q (Qiagen) using the Maxima SYBR Green qPCR Master Mix (ThermoFisher Scientific). GAPDH was selected as a housekeeping gene and data were normalized to its expression. Statistical analysis was performed using REST (relative expression software tool) software. Primer sequences are described in *Table 6*.

## Immunofluorescence analysis

Cells were seeded in four-well plates. Cells were fixed in 4% paraformaldehyde for 15 min, incubated in 0.1 M glycine for 10 min at room temperature (RT), and then in blocking solution composed of 5% goat serum, 0.6% Triton in PBS for 30 min at RT. Cells were immunostained at 4°C overnight in blocking solution with primary antibodies anti-Sox2 (1:300; Millipore), anti-Oct4A (1:400; Cell Signaling), anti-Nanog (1:500; Abcam), anti-SSEA4 (1:50; Millipore), and anti-TRA-1–81 (1:50; Millipore). For the immunofluorescence characterization, samples were incubated with appropriate secondary antibodies Rhodamine-Red anti-mouse IgM and antirabbit IgG, Alexa Fluor 488 anti-mouse IgG, (Jackson ImmunoResearch, distributed by Li StarFish, Milan, Italy) for 1 hr at RT, and nuclei were counterstained with Hoechst 33258. Cells were mounted with GelMount aqueous mounting media (Sigma). The images were acquired using a Leica DMI4000B inverted microscope linked to a DFC360FX or a DFC280 camera (Leica Microsystems).

## Embryoid body and trilineage differentiation

Pluripotency competence of all iPSC clones was assessed by embryoid body formation Presence of the three germ layer derivatives in the generated embryoid bodies was shown in vitro by immunofluorescence staining (*Figure 7—figure supplement 4*). Briefly, cells were thawed and seeded into 6-well plates ($5 \times 10^4$ cells per well) under appropriate culture conditions and incubated at 37°C. At

---

**Table 5.** Molecular karyotype.

aCGH analysis was performed by SurePrint G3 Human CGH Microarray Kit 60K and revealed CNVs on chromosomes 2 and 3 in MH-11 cells. No genomic instability was observed in CPN cells.

| Cell line | Molecular karyotype |
| --- | --- |
| CPN | arr[hg18] (1–22,X)x2 |
| MH-11 | arr[hg18] 2p16.1 (60,966,923–61,003,123)x6, 3q13.31(117,895,555-118,070,129)x7 |

**Table 6.** Primer sequences for qRT-PCR.

| Gene | Forward primer | Reverse primer |
| --- | --- | --- |
| SOX2 | GACAGAGCCCATTTTCTCCA | AAATCCTGTCCTCCCATTCC |
| NANOG | TATGCCTGTGATTTGTGGGC | GTTTGCCTTTGGGACTGGTG |
| OCT4 | AGAGGATCACCCTGGGATAT | CGCCGGTTACAGAACCACAC |
| GAPDH | TGTTCGACAGTCAGCCGCAT | TAAAAGCAGCCCTGGTGACC |

confluency of 80% were detached as cell clumps, plated in six-well low-attachment plates, and cultured in Essential eight media (Thermo Fisher Scientific) supplemented with 4 mg/ml PVA (polyvinyl alcohol, Sigma) and 10 µg/ml ROCK inhibitor. Two days later, cell aggregates were nourished with Essential eight media (Thermo Fisher Scientific) and E6 media (1:1 mixture) supplemented with 4 mg/ml PVA. On day 6, Embryoid Bodies (EBs) were collected, plated on matrigel-coated wells, and allowed to differentiate for further 8 days with daily media changes. For immunofluorescence analysis, on day 14, cells were seeded in four-well plates and fixed in 4% paraformaldehyde, incubated in 0.1 M glycine, and blocked in a solution consisting of 5% goat serum, 0.6% Triton in PBS. Cells were immunostained in blocking solution with primary antibodies: for ectoderm anti-βIII-Tubulin (1:100; Sigma), mesoderm anti- SMA (1:200; Sigma), and endoderm anti-AFP (1:50; R and D Systems). For the immunofluorescence characterization, samples were incubated with appropriate secondary antibodies Alexa Fluor 488 anti-mouse IgG and Rhodamine-Red anti-rabbit IgG (Jackson ImmunoResearch, distributed by Li StarFish, Milan, Italy), and nuclei were counterstained with Hoechst 33258. The images were acquired using a Leica DMI4000B inverted microscope linked to a DFC360FX or a DFC280 camera (Leica Microsystems).

## Induced pluripotent stem cells hematopoietic differentiation

At day −1, colonies of pluripotent stem cells were seeded on Geltrex (12 ug/cm2)-coated 100 mm dish plates, in mTeSR1 (STEMCELL Technologies) with 10 µM ROCK Inhibitor (Millipore). The starting cell concentration was adjusted for each cell line at 10–15% confluency range ($5 \times 10^5$ cells/dish). After 4 hr, media was replaced with fresh mTeSR1. At Day 0, cells were transferred in a xeno-free media based on StemPro-34 SFM (ThermoFischer Scientific), supplemented with Penicillin/Streptomycin 0.5% v/v (Sigma), L-Glutamine 1% v/v (Gibco), 1- Thioglycerol 0.04 mg/mL (Sigma), and ascorbic acid 50 mg/mL (Sigma). This media was retained for the entire experiment and supplemented with different cytokines, small molecules and growth factors, according to the following schedule (*Donada et al., 2019*): days 0–2: BMP4 (10 ng/mL), VEGF (50 ng/mL) and CHIR99021 (2 µM). Days 2–4: BMP4 (10 ng/mL), VEGF (50 ng/mL) and FGF2 (20 ng/mL). Days 4–6: VEGF (15 ng/mL), and FGF2 (5 ng/mL). Day 6: VEGF (50 ng/mL), FGF2 (50 ng/mL), SCF (50 ng/mL), and FLT3L (5 ng/mL). Days 7–10: VEGF (50 ng/mL), FGF2 (50 ng/mL), SCF (50 ng/mL), FLT3L (5 ng/mL), TPO (50 ng/mL), and IL-6 (10 ng/mL). Days 10–14: SCF (50 ng/mL), FLT3L (5 ng/mL), TPO (50 ng/mL), and IL-6 (10 ng/mL). Starting from day 14, CD61$^+$ early megakaryocytic progenitors were sorted by immunomagnetic selection kit (Miltenyi Biotech, Bologna, Italy) and seeded for additional 5 days within the silk bone marrow model in presence of TPO (50 ng/mL) supplemented or not with 500 ng/mL Eltrombopag.

## Evaluation of differentiation and proplatelet formation by ex vivo differentiated megakaryocytes

Megakaryocyte differentiation and proplatelet yields were evaluated by adhesion on fibronectin at the end of the culture (14th day), as previously described (*Di Buduo et al., 2014*; *Pecci et al., 2009*). Briefly, 12 mm glass cover-slips were coated with 25 µg/ml human fibronectin (Merck-Millipore, Milan, Italy), for 24 hr at 4°C. Megakaryocytes were harvested from the silk bone marrow scaffold by extensive washing and seeded in a 24-well plate, at 37°C in a 5% $CO_2$ fully humidified atmosphere. After 16 hr, adhering cells were fixed in 4% paraformaldehyde (PFA), permeabilized with 0.1% Triton X-100 (Sigma Aldrich, Milan, Italy), and stained for immunofluorescence evaluation with rabbit anti-β1-tubulin primary antibody (1:1000) or anti-mouse CD61 (1:100) and Alexa Fluor-conjugated secondary antibodies (1:500) (Invitrogen, Milan, Italy). Nuclei were stained with Hoechst

33258 (1:10,000) (Sigma Aldrich, Milan, Italy). The cover-slips were mounted onto glass slides with ProLong Gold antifade reagent (Invitrogen, Milan, Italy) and imaged by an Olympus BX51 microscope (Olympus, Deutschland GmbH, Hamburg, Germany). Proplatelet-forming megakaryocytes were identified as cells displaying long filamentous structure ending with platelet-sized tips. The results were expressed as a percentage of the total number of cells analyzed.

## Imaging of megakaryocyte cultures within the 3D silk bone marrow model

For immunofluorescence imaging of megakaryocyte cultures within the silk bone marrow tissue model, samples were fixed in 4% paraformaldehyde (PFA) for 20 min and then blocked with 5% bovine serum albumin (BSA, Sigma) for 30 min at room temperature. Samples were probed with anti-CD61 (1:100) overnight at 4°C and then immersed in Alexa Fluor secondary antibody (1:500) for 2 hr at room temperature. Nuclei were stained with Hoechst. Samples were imaged by a TCS SP8 confocal laser scanning microscope (Leica, Heidelberg, Germany). For silk fibroin scaffolds imaging, we took advantage of silk auto-fluorescence in UV light. In some experiments, silk fluorescence was brightened by staining with Hoechst (*Talukdar et al., 2011*). For all immunofluorescence imaging, the acquisition parameters were set on the negative controls. 3D reconstruction and image processing performed using Leica licensed software or Image J software.

## Evaluation of platelet morphology

For analysis of peripheral blood and ex vivo collected platelet morphology, different approaches were used. First, megakaryocytes at the end of differentiation and platelets from peripheral blood or perfused media were visualized by light microscopy with an Olympus IX53 (Olympus Deutschland GmbH, Hamburg, Germany). For analysis of cytoskeleton components, cells were stained as previously described (*Di Buduo et al., 2016*). Briefly, collected platelets were fixed in 4% PFA and centrifuged onto poly-L-lysine coated coverslip while peripheral blood smears were air-dried and then fixed in 4% PFA, permeabilized with 0.1% Triton X-100 for 5 min, and blocked with 5% BSA for 30 min at room temperature. To visualize microtubule organization, samples were probed with anti-β1-tubulin (1:1000) for 1 hr at room temperature and then immersed in Alexa Fluor secondary antibody (1:500) for 2 hr at room temperature. Samples were mounted onto glass slides with ProLong Gold antifade reagent (Invitrogen, Milan, Italy) and then imaged by an Olympus BX51 fluorescence microscope (Olympus, Deutschland GmbH, Hamburg, Germany). For all immunofluorescence imaging, the acquisition parameters were set on the negative controls, which were routinely performed by omitting the primary antibody.

## Flow cytometry

Flow cytometry settings for analysis of megakaryocytes and ex vivo generated platelets were established, as previously described (*Abbonante et al., 2016*; *Cramer et al., 1997*; *Fujimoto et al., 2003*; *Nakamura et al., 2014*; *Takayama et al., 2008*). For analysis of the percentage of fully differentiated megakaryocytes at the end of the culture (14th day), $50 \times 10^3$ cells were suspended in phosphate buffer saline (PBS) and stained with a FITC-conjugated antibody against human CD41 and human CD42b (PE) (eBioscience, Milan, Italy) at room temperature in the dark for 30 min and then analyzed. Ex vivo collected platelets were analyzed using the same forward and side scatters as human peripheral blood and identified as $CD41^+CD42b^+$ events. Isotype controls were used as negative controls to exclude non-specific background signal. The platelet number was calculated using a TruCount bead standard. A minimum of 10,000 events was acquired. All samples were acquired with a Beckman Coulter Navios flow cytometer (Indianapolis, IN, US). Off-line data analysis was performed using Beckman Coulter Navios software package.

## Statistics

Values were expressed as mean plus or minus the standard deviation (mean ± SD) or mean plus or minus the standard error of the mean (mean ± SEM). A two-tailed paired t-test was performed for statistical analysis of data from samples tested in parallel under different experimental conditions. A two-tailed unpaired t-test was performed for statistical analysis of data from different samples.

Statistical analysis was performed with GraphPad Software. A p-value of less than 0.05 or 0.01 was considered statistically significant. All experiments were independently replicated at least three times.

## Acknowledgements

The authors thank Novartis for providing Eltrombopag; 'Centro Grandi Strumenti' of the University of Pavia and Dr. Patrizia Vaghi for technical assistance with confocal microscopy; Prof. Joseph Italiano for providing β1-tubulin antibody. This paper was supported by Cariplo Foundation (Project n. 2017–0920), the US National Institutes of Health (Project n. R01 EB016041-02), Telethon Foundation (Project n. GGP17106), and European Commission H2020-FETOPEN-1-2016-2017-SilkFusion (Project n. 767309).

## Additional information

### Competing interests

James B Bussel: James B Bussel is consultant and participant in advisory boards for Amgen, Novartis, Dova, Rigel, UCB, Argenx, Momenta, Regeneron. The other authors declare that no competing interests exist.

### Funding

| Funder | Grant reference number | Author |
| --- | --- | --- |
| European Commission | H2020-FETOPEN-1-2016-2017-SilkFusion ID 767309 | Alessandra Balduini |
| National Institutes of Health | R01 EB016041-02 | Alessandra Balduini |
| Fondazione Cariplo | 2017-0920 | Christian A Di Buduo |
| Fondazione Telethon | GGP17106 | Alessandro Pecci |

The funders had no role in study design, data collection and interpretation, or the decision to submit the work for publication.

### Author contributions

Christian A Di Buduo, Conceptualization, Data curation, Formal analysis, Funding acquisition, Validation, Investigation, Visualization, Methodology, Writing - original draft; Pierre-Alexandre Laurent, Data curation, Formal analysis, Validation, Investigation, Methodology; Carlo Zaninetti, Resources, Data curation, Formal analysis, Investigation; Larissa Lordier, Data curation, Formal analysis, Validation, Methodology; Paolo M Soprano, Data curation, Validation, Investigation; Aikaterini Ntai, Data curation, Formal analysis, Validation; Serena Barozzi, Resources, Data curation; Alberto La Spada, Data curation, Investigation; Ida Biunno, Data curation, Formal analysis, Investigation, Methodology; Hana Raslova, Resources, Data curation, Formal analysis, Validation, Methodology; James B Bussel, David L Kaplan, Carlo L Balduini, Resources, Validation, Writing - review and editing; Alessandro Pecci, Resources, Formal analysis, Funding acquisition, Validation, Writing - review and editing; Alessandra Balduini, Conceptualization, Resources, Data curation, Formal analysis, Supervision, Funding acquisition, Validation, Investigation, Visualization, Methodology, Writing - original draft, Project administration

### Author ORCIDs

Christian A Di Buduo ⓘD https://orcid.org/0000-0002-6472-2008
Carlo Zaninetti ⓘD http://orcid.org/0000-0003-1754-1260
David L Kaplan ⓘD http://orcid.org/0000-0002-9245-7774
Alessandro Pecci ⓘD http://orcid.org/0000-0001-9202-7013
Alessandra Balduini ⓘD https://orcid.org/0000-0003-3145-1245

## Ethics

Human subjects: Human peripheral blood (PB) samples were obtained from healthy subjects and patients after informed consent prior to participation in the study. All samples were processed following the ethical committee of the local Institution and the principles of the Helsinki Declaration.

## Decision letter and Author response

Decision letter https://doi.org/10.7554/eLife.58775.sa1
Author response https://doi.org/10.7554/eLife.58775.sa2

## Additional files

### Supplementary files

• Transparent reporting form

### Data availability

All data generated or analysed during this study are included in the manuscript and supporting files. Source data files have been provided for: - Figure 2F - Figure 3B, C, D - Figure 3-figure supplement 1 - Figure 4C, E, F - Figure 5B - Figure 6D, E - Figure 7A, C, D - Figure 7-figure supplement 1, - Figure 8D, E, F.

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
