## [Decision Letter]

**Acceptance summary:**

The paper demonstrates a 3D bioreactor that can reliably detect patient-specific defects in iPSC derived megakaryopoiesis and thrombopoiesis. The strengths of the work include the direct comparison of platelet production from peripheral blood progenitors of a cohort of patients whose response to therapy was known, as well as excellent quantitative and imaging data on megakaryopoiesis and thrombogenesis.

**Decision letter after peer review:**

Thank you for submitting your article "Miniaturized 3D bone marrow tissue model to assess response to Thrombopoietin-receptor agonists in patients" for consideration by *eLife*. Your article has been reviewed by 3 peer reviewers, and the evaluation has been overseen by a Reviewing Editor and Jonathan Cooper as the Senior Editor. The following individual involved in review of your submission has agreed to reveal their identity: Diana Passaro (Reviewer #2).

The reviewers have discussed the reviews with one another and the Reviewing Editor has drafted this decision to help you prepare a revised submission.

Summary:

There was a consensus that this in vitro, 3D silk-based megakaryocyte culture system would be of value to grow from peripheral blood MK-derived platelets and help elucidate the pathophysiology of Inherited Thrombocytopenias. There are a number of strengths: the direct comparison of platelet production from peripheral blood progenitors of a cohort of patients whose response to eltrombopag was known, as well as excellent quantitative and imaging data on megakaryopoiesis and thrombogenesis.

However, there was also a consensus that this system is not entirely novel or advantageous from the one described in the authors' 2015 Blood paper. Other weaknesses include 1) reiteration of published data, 2) lack of healthy control comparators for patient data, 3) weak rationale for using an in vitro model to predict patient responses to eltromopag, and 4) single iPSC line derived from a patient whose response to eltrombopag is not provided.

Essential revisions:

1. The silk in vitro marrow microenvironment was already published on by this group in 2015 (Blood) and also in 2017 (Biomaterials) , so Figures 1 and 2 are not novel. The authors should explain the differences between the two systems and explain how the new device would be better suited than their previously described one for this manuscript aims. In what ways is the new system "new and improved"? This explanation should be included in the introduction/discussion. A side-by-side comparison of the chips (original and revised) is encouraged.

2. Please revise and provide the rationale for using both TPO and eltrombopag. Adding data showing the effects of eltrombopag alone would be of interest.

3. Indicate whether this system is predictive of a response to eltrombopag, based on the patients' data. Absolute platelet numbers generated among patients' samples and with TPO plus/minus eltrombopag should be provided.

4. While there was a correlation between Eltrombopag-enhanced thrombopoiesis from patient-derived blood samples and patient response, this correlation was heavily dependent on one outlier with a large response, and a few with a more minimal response. The actual range of platelet enhancement by Eltrombopag mostly ranged from 3-9 fold. Also, the model is not so reliable that a dose of Eltrombopag should not be tested in a patient based on the in vitro finding, so the true predictive value of such cultures would hopefully be applicable to something more expensive or potentially toxic than simply testing the patient's response to Eltrombopag. In all of the 3D culture studies, TPO and Eltrombopag are given together. What are the effects of Eltrombopag alone? It seems that patients only receive Eltrombopag. What is the rationale of treating with combined TPO and Eltrombopag in vitro?

---

## [Author Response]

Summary:There was a consensus that this in vitro, 3D silk-based megakaryocyte culture system would be of value to grow from peripheral blood MK-derived platelets and help elucidate the pathophysiology of inherited thrombocytopenias. There are a number of strengths: the direct comparison of platelet production from peripheral blood progenitors of a cohort of patients whose response to eltrombopag was known, as well as excellent quantitative and imaging data on megakaryopoiesis and thrombogenesis.However, there was also a consensus that this system is not entirely novel or advantageous from the one described in the authors' 2015 Blood paper. Other weaknesses include 1) reiteration of published data, 2) lack of healthy control comparators for patient data, 3) weak rationale for using an in vitro model to predict patient responses to eltromopag, and 4) single iPSC line derived from a patient whose response to eltrombopag is not provided.

We thank the Reviewers for the generally positive comments and, most importantly, for their thorough efforts to identify areas for improvement in our work. In the revised manuscript we support the novelty of our model compared to existing silk-based technologies as it is a simplified bone marrow scaffold for screening individual responses to drug regimens ex vivo, requiring a much smaller volume of patient’s blood to study platelet production. The highly novel validation of the system, gained by comparing results obtained ex vivo with data from in vivo treatment of the same patients, represents a proof-of-concept for the future applicability of the experimental approach to a wide variety of new compounds intended to treat bone marrow diseases. Data from healthy controls have been provided, and the added value of including iPSCs in the system has been offered in Discussion.

Essential revisions:1. The silk in vitro marrow microenvironment was already published on by this group in 2015 (Blood) and also in 2017 (Biomaterials) , so Figures 1 and 2 are not novel. The authors should explain the differences between the two systems and explain how the new device would be better suited than their previously described one for this manuscript aims. In what ways is the new system "new and improved"? This explanation should be included in the introduction/discussion. A side-by-side comparison of the chips (original and revised) is encouraged.

We thank the Reviewers for this comment. In the past five years, our expertise in modeling megakaryopoiesis and silk processing has led to the development of early models of silk-bone marrow able to produce millions of human platelets from cord blood-derived haematopoietic stem and progenitor cells with the goal of providing proof-of-concept evidence of their applicability for transfusion medicine. We believe that our new miniaturized bone marrow represents a substantial innovation over these models as this simplified system can be easily applied to drug testing starting from patients’ peripheral blood progenitors. Major advantages of the miniaturized model include: (i) rapid manufacture and customization, (ii) easy handling, (iii) the implementation of small silk-based three-dimensional scaffolds to allow the seeding of a small number of adult megakaryocyte progenitors (such as those that can be obtained from 10-15 ml of patients’ peripheral blood), and (iv) the manufacturing of bioreactors holding two identical silk scaffolds which are cultured and perfused in parallel for the direct comparison of two samples.

Our first silk-based bone marrow model (Di Buduo et al., 2015) was composed of a porous silk tube, mimicking vessels, surrounded by a 3D silk matrix to allow the recording of megakaryocyte functions (i.e., migration, proplatelet formation) in response to variations in extracellular matrix components, surface topography and stiffness, and co-culture with endothelial cells. Millions of human platelets were produced and shown to be functional based on multiple activation tests. This system demonstrated the fundamental qualities of silk fibroin for studying thrombopoiesis, such as non-thrombogenicity and the possibility to entrap different molecules while retaining their bioactivity. A limitation of this model was the use of custom-made chambers and silk tubes whose production could not be standardized or scaled easily to guarantee the reliable comparison of different parallel culture conditions. This has been made possible by our miniaturized model. The miniaturized model can produce measurable numbers of platelets starting from less than one-fifth of the cells seeded in the original silk bioreactor, making it possible to study platelet production starting from progenitors collected from 15 ml of patient and control peripheral blood.

Our second silk-bone marrow model (Di Buduo et al., 2017) consisted of a scaled-up version intended to house a larger number of mature megakaryocytes producing millions of platelets for functional studies and biochemical characterization. The flow chamber was made of polydimethylsiloxane, holding a silk sponge, prepared with salt leaching methods, and functionalized with extracellular matrix components. Perfusion of the chamber allowed the recovery of platelets when the silk sponge was cultured with cord blood-derived megakaryocytes.

As indicated above, the major advantages of the present miniaturized bone marrow model over the previous ones are both the requirement for seeding an order of magnitude less megakaryocytes and the standardized production process which is easily scalable to produce as many chambers as needed within the same device. In future applications, we plan to use the system to test more than two drugs at the same time to help clinicians verify the best therapeutic approach for individual patients and to determine critical factors related to clinical platelet output. The miniaturized bone marrow model may pave the road to personalized medicine in this field – an aim and an application that could not be pursued with the previous silk-based bone marrow models.

A summary of key points is now reported in the section Discussion of the revised manuscript. Please find Author response table 1 reporting a side-by-side comparison of the devices:

**Author response table 1. resptable1:** 

	Blood 2015	Biomaterials 2017	Current Manuscript
Size of the cell-seeding well	15x20x5 mm (1500mm^3^)	3.5x20x5 mm (350mm^3^)	2x15x3.5 mm (105mm^3^)
No. of chambers that can be perfused in parallel	Max. 2	Max. 1	> 2 (up to at least 4)
Source of blood	Human Umbilical Cord Blood	Human Umbilical Cord Blood	Human Peripheral Blood
Type of haemopoietic stem and progenitor cells	CD34^+^	CD34^+^	CD45^+^CD34^+^-derived iPSCs
Type of cells seeded	CD34^+^-derived megakaryocytes	CD34^+^-derived megakaryocytes	CD45^+^-derived megakaryocyte progenitorsiPSC-derived megakaryocyte progenitors
No. of cells seeded	2.5x10^5^	4x10^5^	5x10^4^
Time of the culture	24 hours	24 hours	> 7 days
Major application	Studying mechanisms of thrombopoiesis	Producing a high number of platelets	Drug Testing in individual patients

2. Please revise and provide the rationale for using both TPO and eltrombopag. Adding data showing the effects of eltrombopag alone would be of interest.

We thank the Reviewers for the comment on this key point of the experimental design that was not described sufficiently in the previous version of the manuscript.

With the only exception of congenital amegakaryocytic thrombocytopenia, subjects with Inherited Thrombocytopenias have normal or moderately increased serum levels of endogenous thrombopoietin. in vivo, both Eltrombopag and endogenous thrombopoietin are present in the plasma of patients undergoing Eltrombopag treatment. We have shown that serum thrombopoietin concentration was moderately increased, compared to healthy subjects, both in patients with *MYH9*-RD and in those with *ANKRD26*-RT, both before and during Eltrombopag administration (Zaninetti et al., 2020). Combining Eltrombopag and thrombopoietin in the 3D system intends to mimic what patients’ cells experience in vivo in their native environment during Eltrombopag treatment, when bone marrow haemopoietic stem/progenitor cells and megakaryocytes are exposed to stimuli from both molecules. This condition was compared to thrombopoietin alone as a control reference, reproducing the in vivo situation in which the patients are not under treatment with Eltrombopag.

A stimulatory effect of Eltrombopag on stem cell self-renewal has been demonstrated independently of the thrombopoietin receptor (Kao et al., 2018). It is well known that Eltrombopag and thrombopoietin do not compete for the binding to the thrombopoietin receptor due to their affinity for different binding sites but, rather, Eltrombopag has an additive effect on thrombopoietin (Kuter, 2013).

Based on this knowledge, the revised paper contains a detailed description of the rationale supporting our choice to test Eltrombopag in combination with recombinant human thrombopoietin.

3. Indicate whether this system is predictive of a response to eltrombopag, based on the patients' data. Absolute platelet numbers generated among patients' samples and with TPO {plus minus} eltrombopag should be provided.

We thank the Reviewers for this suggestion that greatly improves the reported evidence of the clinical relevance of our findings. The revised version of the manuscript includes a table summarizing platelet counts of investigated patients before Eltrombopag treatment; increase in platelet counts obtained in vivo with Eltrombopag administration; numbers of platelets produced ex vivo with thrombopoietin alone; increase in the numbers of platelets produced ex vivo with the addition of Eltrombopag (Table 2). The significant correlation between platelet count increase ex vivo and in vivo is reported in Figure 6D.

4. While there was a correlation between Eltrombopag-enhanced thrombopoiesis from patient-derived blood samples and patient response, this correlation was heavily dependent on one outlier with a large response, and a few with a more minimal response. The actual range of platelet enhancement by Eltrombopag mostly ranged from 3-9 fold. Also, the model is not so reliable that a dose of Eltrombopag should not be tested in a patient based on the in vitro finding, so the true predictive value of such cultures would hopefully be applicable to something more expensive or potentially toxic than simply testing the patient's response to Eltrombopag. In all of the 3D culture studies, TPO and Eltrombopag are given together. What are the effects of Eltrombopag alone? It seems that patients only receive Eltrombopag. What is the rationale of treating with combined TPO and Eltrombopag in vitro?

The range of variability of the in vitro responses, including the presence of an outlier with a very marked increase of platelet count and some patients with minimal responses, reflects the still unexplained variability of the in vivo clinical response to Eltrombopag. The availability of blood samples and clinical data from the same patients treated with Eltrombopag in vivo offered the unique possibility to validate the miniaturized bone marrow model. The concordance between in vivo and ex vivo data served as proof of the reliability of our laboratory assessment and the broader applicability of the experimental approach. As reported in the ‘*Discussion’* of the revised manuscript, we agree with the Reviewers that, based on these results, such a system will represent a benchmark for pre-clinical testing of other potential pharmacologic treatments (e.g.; new drugs, expensive therapies) intended to cure Inherited Thrombocytopenias or other hematologic diseases affecting blood cell progenitors as well as for testing the effects of potentially toxic agents for the whole haematopoietic niche (e.g.; chemotherapy). The miniaturized bone marrow model will serve as a tool for answering open questions related to mechanisms responsible for different individual responses to thrombopoietin-receptor agonists observed in patients affected by the same disease.

Regarding the rationale for treating samples with combined thrombopoietin and Eltrombopag, please refer to response to the Essential Revision 2.